

# The role of emission reductions and the meteorological situation for air quality improvements during the COVID-19 lockdown period in Central Europe

Volker Matthias[1], Markus Quante[1], Jan A. Arndt[1], Ronny Badeke[1], Lea Fink[1], Ronny Petrik[1], Josefine Feldner[1], Daniel Schwarzkopf[1], Eliza-Maria Link[1], Martin O.P. Ramacher[1], Ralf Wedemann[1]

[1]Helmholtz-Zentrum Hereon, Max-Planck-Straße 1, 21502 Geesthacht, Germany

*Correspondence to:* Volker Matthias (volker.matthias@hereon.de)

**Abstract.** The lockdown measures taken to prevent a rapid spreading of the Corona virus in Europe in spring 2020 led to large emission reductions, particularly in road traffic and aviation. Atmospheric concentrations of $NO_2$ and $PM_{2.5}$ were mostly reduced when compared to observations taken for the same time period in previous years, however, concentration reductions may not only be caused by emission reductions but also by specific weather situations.

In order to identify the role of emission reductions and the meteorological situation for air quality improvements in Central Europe, the meteorology chemistry transport model system COSMO-CLM/CMAQ was applied to Europe for the period 1 January to 30 June 2020. Emission data for 2020 was extrapolated from most recent reported emission data and lockdown adjustment factors were computed from reported activity data changes, e.g. google mobility reports. Meteorological factors were investigated through additional simulations with meteorological data from previous years.

The results showed that lockdown effects varied significantly among countries and were most prominent for $NO_2$ concentrations in urban areas with two-weeks-average reductions up to 55% in the second half of March. Ozone concentrations were less strongly influenced (up to +/- 15%) and showed both, increasing and decreasing concentrations due to lockdown measures. This depended strongly on the meteorological situation and on the NOx/VOC emission ratio. $PM_{2.5}$ revealed 2-12% reductions of two-weeks-average concentrations in March and April, which is much less than a different weather situation could cause. Unusually low $PM_{2.5}$ concentrations as observed in Northern Central Europe were only marginally caused by lockdown effects.

The lockdown can be seen as a big experiment about air quality improvements that can be achieved through drastic traffic emission reductions. From this investigation, it can be concluded that $NO_2$ concentrations can be largely reduced, but effects on annual average values are small when the measures last only a few weeks. Secondary pollutants like ozone and $PM_{2.5}$ depend more strongly on weather conditions and show a limited response to emission changes in single sectors.

## 1 Introduction

The global spread of the Corona virus since the start of 2020 resulted in unprecedented emission reductions caused by lockdown measures in many parts of the world. In Europe, significant reductions in road and air traffic as well as in industrial activities began between end of February and mid of March 2020. Emissions were heavily reduced



in short time, but then steadily increased again as lockdown measures were lifted step by step, until they reached
approximately previous year levels in summer (Forster et al., 2020). However, this temporal emission behaviour
varied from country to country and among the different emission sectors. Emission reductions between the second
half of March and end of June 2020 were probably the largest in Europe since decades, in particular in traffic.
From an air quality perspective, this can be regarded as a huge real world experiment about the effects of severe
emission reductions on air pollutant concentrations and possible side effects of emission reduction measures, e.g.
on secondary pollution formation.
Observational data at ground level and from satellite showed large, but regionally different reductions in $NO_2$
concentrations (e.g. Bauwens et al. (2020);Menut et al. (2020);Velders et al. (2021);Lonati and Riva (2021). For
particulate matter (PM), concentration reductions were less clear and not necessarily in line with the expectations
that would follow the estimated emission reductions. Obviously, also weather conditions have a significant impact
on pollutant concentration levels, but despite the high number of publications that analyse COVID-19 lockdown
effects on air pollution, meteorological influences are mostly not taken into account properly (Gkatzelis et al.,
2021). Wind direction determines strongly the advection of gases and aerosols from distant regions into the area
of interest, higher wind speeds can activate additional emission sources like re-suspension of deposited particles,
and precipitation amounts control deposition. In Central Europe, a period between mid of March and mid of April
was very sunny and dry, both conditions that favour the formation of secondary pollutants like ozone and PM and
that hamper particle deposition. On the other hand, advection of clean air from northern Europe influenced
pollution levels in northern Central Europe in the beginning of April, as well.
As has been pointed out in recent publications about the effect of COVID lockdown emission reductions on air
pollutant concentrations (e.g. Menut et al. (2020);Velders et al. (2021)), the relationship between emissions and
concentrations is not necessarily straightforward and easy to explain. A simple comparison between before and
after lockdown concentrations neglects seasonal and weather effects. A similar argument holds for comparisons
with the same week of the previous year. While seasonal effects are considered in this case, the weather situation
might still be very different. In addition, technology or economically driven emission changes from one year to
another are not taken into account. Chemistry transport models and sophisticated emission models can help in
disentangling the relationships between emissions, meteorology, and concentration levels. In addition, they can
quantify the contribution of different source sectors and investigate effects of reduced concentrations of specific
pollutants on the formation of other secondary species. For example, it has been discussed by Kroll et al. (2020)
and (Huang et al., 2020) that lower NO emissions might lead to higher ozone concentrations and a higher potential
for the oxidation of organics, which might result in increased secondary organic aerosol (SOA) formation. In fact,
Amouei Torkmahalleh et al. (2021) analysed observed $NO_2$ and $O_3$ concentrations in numerous cities around the
world and report increased ozone in urban environments. However, depending on the NOx/VOC emission ratios
and the meteorological situation, the effects might differ from place to place (see e.g. Mertens et al. (2021)).
To quantify the effects of the lockdown measure on ambient concentrations, these need to be separated from other
sources of influence which predominantly are assumed to be the meteorological conditions. For Europe, Menut et
al. (2020) assessed the influence of lockdown measures on air quality without the biases of meteorological
conditions in an ad-hoc modelling study for March 2020. They compared a reference model run with 2017
emission data for Europe to a lockdown run with estimated emission reductions. Both runs were based on the
same meteorological fields. Decreases in $NO_2$ concentrations ranging from −30% to −50% in all western European





countries due to the lockdown measures alone have been found. The effect on fine particle concentrations has
been comparably less pronounced (−5 to −15%). Sharma et al. (2020) performed a similar study for India. Around
43%, 31%, 10%, and 18% decreases in $PM_{2.5}$ , $PM_{10}$, CO, and $NO_2$ in India were observed during the lockdown
period compared to previous years. While, there were 17% increase in $O_3$ and negligible changes in $SO_2$. With
focus on the Netherlands, Velders et al. (2021) used a machine learning (ML) algorithm (Random forest) to
remove the effects due to meteorological variability on pollutant concentrations. Concentrations that were
measured before and during the lockdown period are compared with the "expected" concentrations during this
period, according to the ML algorithm and the differences are ascribed to the lockdown measures. The authors
also applied chemical transport modelling to assess the question of separating the effects. They concluded that the
unusual 2020 meteorology in the Netherlands led to decreased $PM_{10}$ and $PM_{2.5}$ concentrations by about 8% and
10%, respectively, but the NOx, $NO_2$, and $O_3$ concentrations were not affected. In a study addressing the air quality
during the lockdown period in Milan (Collivignarelli et al., 2020) used a different procedure based on
observations, only, aiming to eliminate the influence of weather phenomena on the air quality. To do so, they
identified a meteorological reference period in the same year around the lockdown phase. About two weeks in
February (7th to 20th) were considered suitable to serve as a control time segment, for which gas and particle
concentrations were used to quantify the lockdown effects. Using machine-learning (ML) models fed by
meteorological data along with other time features. Petetin et al. (2020) estimate the $NO_2$ mixing ratios for Spain
that would have been observed in the absence of the lockdown. So-called meteorology-normalized $NO_2$ reductions
induced by the lockdown measures were quantified by comparing the estimated business-as-usual values with the
observed $NO_2$ mixing ratios. It was found that the lockdown measures were responsible for a 50% reduction in
$NO_2$ levels on average over all Spanish provinces and islands during the period from 14 March to 23 April 2020.
Additionally, van Heerwaarden et al. (2021) used ground based and satellite observations in combination with
radiative transfer modelling to disentangle meteorological effects and those of aerosol emission reduction and
reduced contrails on observed record irradiance in Western Europe. They concluded that lockdown measures were
far less important for the irradiance record than the exceptionally dry and particularly cloud-free weather.
In this paper we present results derived with the COSMO-CLM/CMAQ model system together with a highly
modular emission model to quantify the contribution of the estimated emission reductions on the concentrations
of $NO_2$, $O_3$ and $PM_{2.5}$ in Central Europe and to separate the contribution of emission changes from those caused
by distinct weather patterns. CMAQ was fed with updated emission data for the year 2020, including time profiles
for sectors and countries that approximate the lockdown emission reductions. Chemistry transport model
simulations were performed for January – June 2020. The effects of distinct weather patterns on the effects of
emission reductions on pollutant concentrations were investigated through additional simulations with
meteorological conditions for the same time period in recent previous years with very different weather conditions.
The results allow for an interpretation of the observed concentration reductions when compared to previous years.
It also gives a range of possible concentration changes resulting from the same emission reductions.

## 2 Model simulations

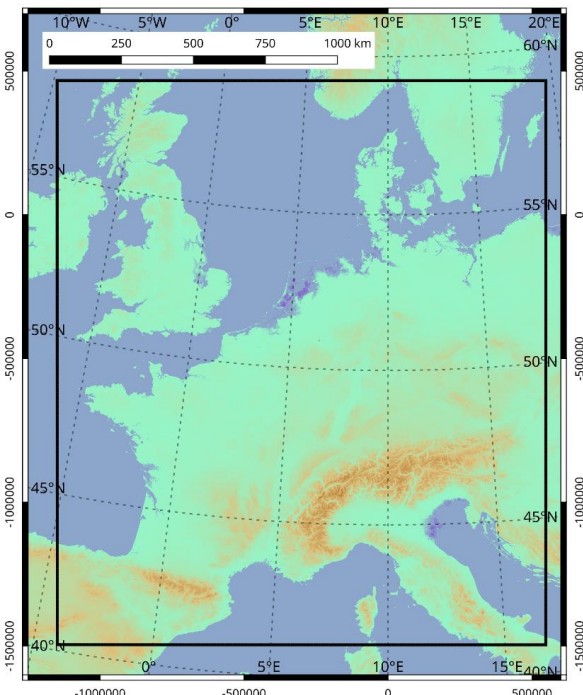

**Figure 1: Inner domain of the CMAQ model (black line) along with the coordinates of the CMAQ projection (values outside the zebra frame)**

This study focuses on the effects of emission reductions during the lockdown in Central Europe in spring and early summer 2020. While emission changes were considered for entire Europe, the main area under investigation w.r.t. effects on concentrations covers the most populated regions in Central Europe (Fig .1), only. This restriction was applied for the sake of a higher resolution and for allowing a reasonable interpretation of meteorological impacts. The Community Multi-scale Air Quality Model (CMAQ) (Byun and Schere, 2006;Byun and Ching, 1999) version 5.2 was used with the carbon bond 5 (CB05) photochemical mechanism (CB05tucl) (Kelly et al., 2010)and the AE6 aerosol mechanism. The model was run for 2020 with a spin-up time of 2 weeks in 2019 to avoid the influence of initial conditions on the modelled atmospheric concentrations. CMAQ was set up on a 36 x 36 km2 grid for entire Europe and for a one-way nested 9 x 9 km2 grid for Central Europe, see Fig. 1. The vertical model extent comprises 30 layers from the model surface up to the 100 hPa pressure level. Twenty of these layers are below approx. 2000 m, and the lowest layer has a height of 36 m.

Chemical boundary conditions for the outer model domain were taken from the IFS-CAMS analysis (Inness et al., 2019b) available from the MARS archive at ECMWF and the Copernicus Atmosphere Monitoring Service Atmosphere Data Store (Inness et al., 2019a). Particle and gas concentration fields of the Global Analysis and Forecast are provided on a T511 spectral grid with 137 vertical levels. The IFS-CAMS data were temporally and spatially remapped onto the boundary of the CMAQ domain. Finally, a unit conversion and a transformation of the chemical species from IFS-CAMS to CMAQ were applied.





Meteorological data for the CMAQ model were provided by a simulation of the COSMO model (Baldauf et al.,
2011;Doms et al., 2011;Doms and Schättler, 2002) applying the version COSMO5-CLM16 (climate mode
(Rockel et al., 2008)). To simulate the radiative transfer as realistic as possible, an extension of the COSMO model
for the MACv2 transient aerosol climatology was used. The soil was initialized taking the data from a 40 year
simulation with the COSMO model. Then, the atmospheric simulations were performed for the period 1
September 2019 to 30 June 2020 using the MERRA2 Global reanalysis (Gelaro et al., 2017) as initial and lateral
boundary conditions. The same was done for the periods 1 September 2015 to 30 June 2016 and 1 Sep 2017 to 30
June 2018. To ensure that the atmospheric fields in the transient model integration are close to the observations
over the whole period of 10 months, a nudging technique was used as described in Petrik et al. (2021). The reader
is referred to this publication to find more information about the setup of the atmospheric model (setup 'CCLM-
oF-SN').
CMAQ simulations were performed with emissions as they could be expected for 2020 without any lockdown
measures and with another emission data set that was modified according to reported changes in traffic and
industrial activities. The latter is regarded as the emission data set that reproduces real world emissions during the
first COVID-19 lockdown phase in 2020 best. In the following we will refer to this simulation as the COV case,
while the simulations with expected emissions without lockdown is referred to as the noCOV case. The difference
between the simulated pollutant concentrations for the two cases represents the COVID-19 lockdown effects on
air quality. A detailed description of the emission data construction is given in the next section. Additional model
simulations with meteorological conditions for the years 2016 and 2018 have been performed with CMAQ using
the same 2020 emission data sets.
**3 Emission data**
**3.1 Basic emissions 2020, noCOV case**
Emissions are based on the CAMS-REGAP-EU version 3.1 available at the ECCAD website
(https://permalink.aeris-data.fr/CAMS-REG-AP). The dataset comprises annual totals for anthropogenic
emissions in 13 GNFR sectors (Granier et al., 2019). The most recent data set was for 2016. For this study, the
emission data was extrapolated to the year 2020 based on the temporal emission development in previous years.
For the application in the CMAQ model the data was re-gridded and vertically and temporally redistributed.
Additionally, in order to investigate the effects of lockdown measures on the emissions, sector and country specific
temporal profiles of lockdown effects were applied. The data preparation was done with a modular toolbox for
emission calculation, the Highly Modular Emission MOdel (HiMEMO), currently developed at Helmholtz-
Zentrum Hereon. The framework is built in the R programming language, using the libraries netcdf, proj4, sp,
raster and their dependencies.
HiMEMO was run with gridded emission data from the CAMS inventory for 2016 in a spatial resolution of 0.05°
x 0.1°. The inventory contains gridded annual emissions for chemical species groups, i.e. NOx, NMVOC, CO,
NH$_3$, CH$_4$, SO$_2$, PM$_{2.5}$ and PM$_{10}$. Several of these chemical groups need to be split into chemical components, or
sub-groups of species according to the CB05 chemical mechanism used by CMAQ. The NOx split was done by
applying a NO/NO$_2$ ratio of 90/10 for traffic, a ratio of 92/8 for shipping and 95/5 for all other sectors. Land based
NMVOC emissions were split for individual sectors. PM was split as described by Bieser et al. (2010) for the



SMOKE for Europe emission model. All other species in the CAMS-REGAP-EP inventory were directly
transferred to CMAQ.
Vertical emission distributions per sector follow Bieser et al. (2011). The vertical distribution for the shipping
sector was treated differently for land and ocean-going ships, the latter being emitted in altitudes up to 100 m. The
temporal profiles follow those provided by TNO (Denier van der Gon et al. (2011), also described in Matthias et
al. (2018))
Biogenic emissions of VOCs (BVOCs) and NO were calculated with the Model of Emissions of Gases and
Aerosols from Nature (MEGAN) (Guenther et al., 2012). Version 3 of MEGAN (Guenther et al., 2020) was used
in this study, it was driven by preprocessed meteorological data for CMAQ as described above. Vegetation data
tables were downloaded from the MEGAN website and not further modified for this study. Leaf area index (LAI)
data was taken from GEOV1 products (SPOT/PROBA V LAI1) as an alternative input for MEGAN3 (Baret et
al., 2013).
The annual data for 2016 were extrapolated to 2020 for each national emission sector according to the Gridded
Nomenclature For Reporting (GNFR) in order to produce expected emissions for 2020 without lockdown effects.
The starting point were the time series data of yearly totals for the pollutants BC, CO, NH3, NMVOC, NOx, $PM_{10}$,
$PM_{2.5}$ and $SO_2$, which are provided by the EMEP centre on emission inventories and projections (EMEP/CEIP
2020 Present state of emission data; https://www.ceip.at/webdab-emission-database/reported-emissiondata).
Using the time series data a mean annual change rate for emissions (CE, in %) was derived for each pollutant,
sector and country, separately. The projection of the 2016 emissions to the year 2020 was realized through a
projection factor PF=1+ CE/100*(2020-2016). Using a mean change rate based on the development of emissions
within the 3 years 2017-2019 (method 1), PF could be very large (more than 2) for some countries and sectors.
This can result from large changes and fluctuating time series of the yearly emissions. In order to avoid very large
and presumably erroneous emission changes between 2016 and 2020, a maximum allowed annual change rate
was introduced. If the CE was larger than 10%, a modified CE was computed by considering the entire time series
of annual emissions, but not more than ten years (method 2). If there still was a CE of more than 10%, we limited
it to a maximum change of ±10%. Regarding the shipping sector, no changes were assumed between the years
2016 and 2020.
**3.2 Lockdown effects, COV case**
For the lockdown scenario, we adjusted national emissions from the following GNFR sectors: A_PublicPower,
B_Industry, F_RoadTransport, G_Shipping and H_Aviation. Lockdown emission reduction functions, here called
Lockdown Adjustment Factors (LAF) were calculated based on published data sources that resemble the effects
of lockdown measures on a daily basis. LAFs were derived for 42 European countries and two sea basins, the
North Sea and the Baltic Sea.
The datasets used for the construction of the modification functions are described in the following. If the input
data was not available for an individual country, data from a neighbouring country was used to estimate the
reduction. A table showing the data availability per sector and country is given in the appendix (Table A1). The
modification functions are applied to all species, heights and time steps of the anthropogenic emission dataset for

207   2020.

**A_PublicPower and B_Industry**





Eurostat data (https://ec.europa.eu/eurostat/databrowser/view/sts_inpr_m/default/bar?lang=en) was used to
account for changes in the sectors A_PublicPower and B_Industry.
The energy data provided there comprise monthly information on the volume index of production for electricity,
gas, steam and air conditioning supply. They are available for 35 countries in Europe. The industry data comprise
monthly information on the volume index of production for mining and quarrying; manufacturing; electricity, gas,
steam and air conditioning supply and construction and are available for 20 countries in Europe. The indices are
based on an index value of 2015. However, since we want to use them to evaluate the lockdown period, we
normalized the changes based on the January 2020 value. The data are given in a monthly resolution, however,
for many countries in Europe the lockdown started in mid of March. Therefore, a piecewise cubic spline
interpolation procedure was applied to derive daily lockdown adjustment factors while still maintaining the
monthly values. Examples are given for both sectors in Germany in Fig. 2.

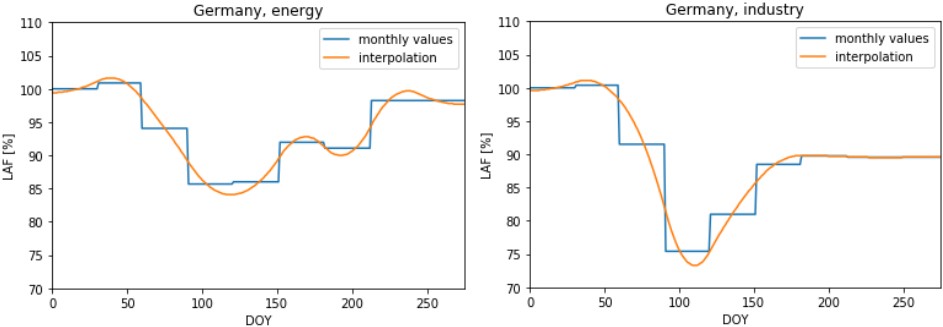


**Fig. 2: Examples for monthly values and interpolated functions for Lockdown Adjustment Factors (in %) for the**
**sectors A_PublicPower and B_Industry in Germany.**
**F_RoadTransport**
Google Mobility Reports (https://www.google.com/covid19/mobility/) deliver daily percentage change of visits
in different areas (e.g. residential, transit, recreation, work places). The reference value is the median of the
corresponding weekday between 3rd of January and 6th of February 2020. We use Google Mobility Reports for
transit on a national level to account for the changes in road traffic emissions. Through this method, reduced traffic
on national holidays, e.g. around Easter and 1 May are considered as well.
**G_Shipping**
To derive scaling factors that account for ship traffic and emission reductions in this sector, bottom-up ship
emission inventories were created with the HiMOSES ship emission model (Schwarzkopf et al., 2021) using
Automatic Identification System (AIS) data for 2019 and 2020 covering the German Bight and the Western Baltic
Sea. The data was recorded in Bremerhaven and Kiel by the German Federal Maritime and Hydrographic Agency
(BSH) .A 7-days rolling mean filter was applied to the calculated $CO_2$ emission ratios (Figure 3). On average, the
data revealed a slight reduction of ship traffic in the North Sea area by approx. 10%. For the Baltic Sea traffic
reductions were clearly visible with a downward trend from March until mid of June that could be mainly
attributed to RoRo and passenger ships. For the first 75 days of the year until 15 March 2020 no reductions were





applied, afterwards daily LAF were used similar to the approach for road traffic. LAFs for the North Sea were
also applied for the Mediterranean Sea, those for the Baltic Sea were also applied to inland shipping. The reasoning
behind this is that shipping in the Mediterranean is mostly international cargo transport, similar to the North Sea,
and inland navigation is connected to short range transport, similar to the Baltic Sea. As can be seen in Fig.3
relative increases in shipping emissions might also occur during limited time.

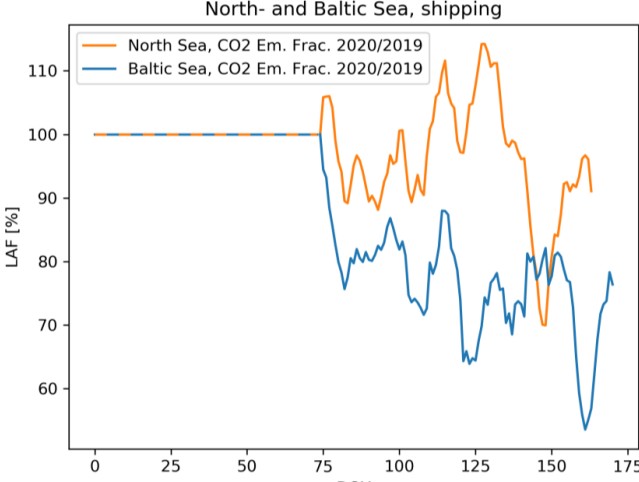


**Fig 3: Lockdown adjustment factors created from the seven days rolling mean ratios of CO2 emissions from shipping**
**in 2020 relative to 2019. Until day 75 (15 March) no changes and a LAF of 1 was assumed.**

**H_Aviation**
Airport traffic total arrivals and departures data from Eurocontrol (https://ansperformance.eu/data) were used to
account for emission changes in the aviation sector. We applied a reduction based on a weekday mean from 3
January 2020 until 6 February 2020, similar to Google mobility data. Daily values for 42 European countries are
available. The relative reductions in this sector were most pronounced, reaching -90% in March and April and a
slower recovery than the other sectors.



**Sector Comparison**

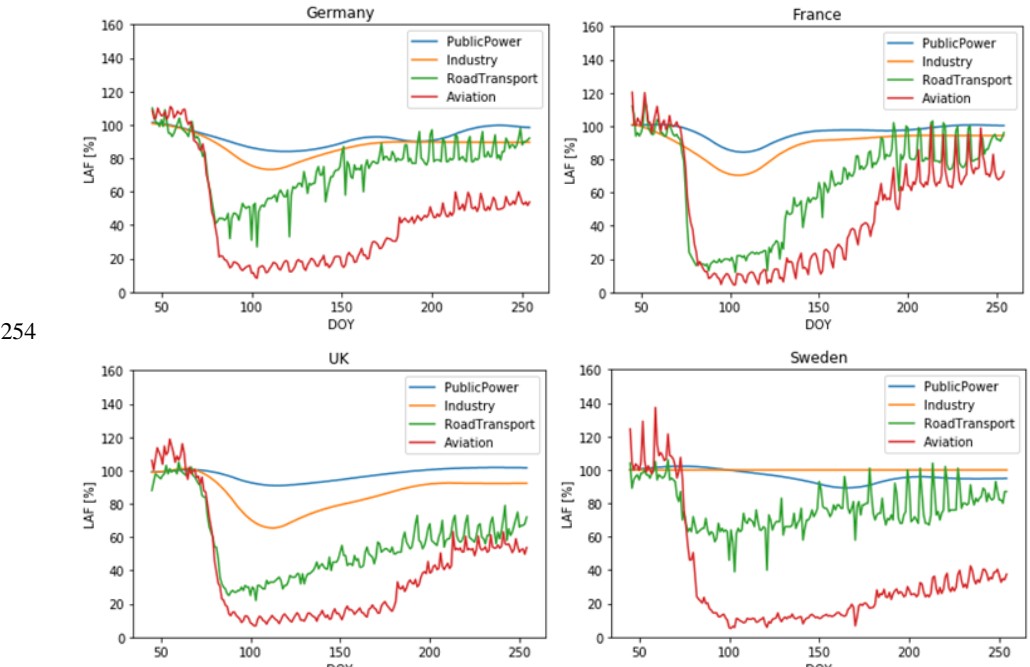



**Fig. 4: LAFs for Germany (a), France (b), United Kingdom (c) and Sweden (d) for the sectors: A_PublicPower,**
**B_Industry, F_RoadTransport, and H_Aviation**
LAFs for Germany, France, UK and Sweden are exemplarily shown in Figure 4. Huge emission reductions in
road traffic and air traffic between 10 and 20 March can clearly be seen. Public power and industry, on the other
hand, show much smaller reductions (10-30%) and almost reach previous year levels until the end of June. At the
same time in France and Germany, road traffic was back to 90% of the previous year, however in the UK and in
Sweden 20-40% reductions were still visible in the activity data. . Comparisons of country-specific LAFs for the
sectors F_RoadTransport, and H_Aviation are given in the supplement (Fig. A1 and A2).
Figure 5 presents total daily NOx emissions in the entire Central European domain (see Fig. 1) for the time period
from 1 January to 30 June 2020 for the COV and the noCOV case separated by GNFR sectors. Road transport is
the most important emission sector with approx. 20 to 30 %, followed by ocean shipping, other stationary
combustion, industry and public power, which all have similar contributions of approx. 10 %. Combustion shows
a clear decline towards the summer months due to the fact that domestic heating is mainly necessary in winter.
Reductions caused by the lockdown stem mostly from the road transport sector, with a strong drop in emissions
starting around day 75 (15 March). The aviation sector, which experienced the strongest relative drop in emissions
during the lockdown, does not play a major role for the overall emission of NOx. However, it might be important
near airports and in the upper troposphere. Overall, NOx emissions in Central Europe dropped by around 25000
mol/s (approx. 4 kt/h, when given as $NO_2$) during the strictest lockdown period in late March and early April. This
corresponds to a relative drop of around -30% (Fig. 5).





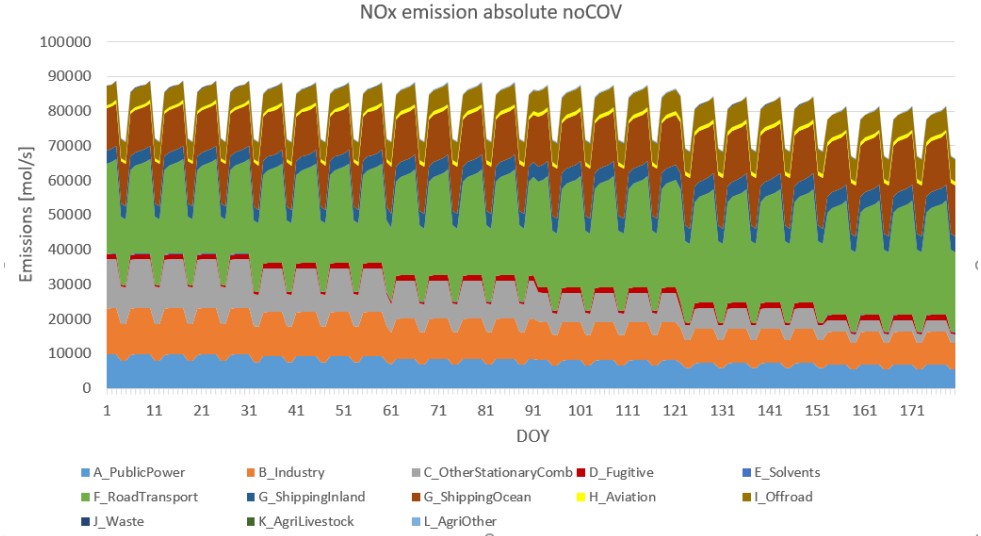


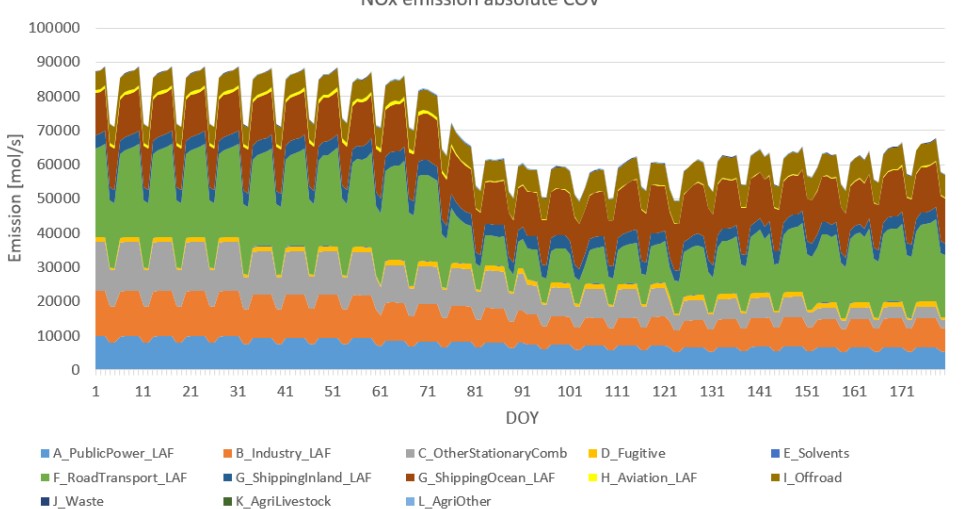


**Fig.5: Daily average values for sector separated NOx emissions summarized over the entire Central European model**
**domain for the noCOV and the COV case (with LAF).**
**4 Observational data**
We focus our analysis on the most important air pollutants for human health, namely $NO_2$, $O_3$ and $PM_{2.5}$. In this
chapter, first the meteorological situation between 1 January and 30 June 2020 is analysed. Afterwards,
observational air quality data at six selected measurement stations within the EEA network
(https://www.eionet.europa.eu/countries/index) are presented and discussed.





### 4.1 Meteorological situation

During the lockdown period in spring 2020 large parts of the region of interest experienced exceptional weather, what is assumed to have a strong influence on concentrations of some of the pollutants in focus.

The weather conditions during the first half of the year 2020 show strong variations across the months and a different character in the northern part of our model domain compared to more southern regions like the Po Valley. While in the North February was extremely wet and windy (south-westerly direction), the second half of March and April were very dry and sunny. Thus for meteorological reasons a comparison of pre-lockdown pollutant concentrations with those during the lockdown is fairly meaningless in assessing the effect of corona measures on the concentrations in the central and northern part of the region of interest. This appears to be different for some more southerly areas, e.g. Collivignarelli et al. (2020) identified a 14 day period in February 2020 for Milan, which they could use as pre-lockdown reference to evaluate emission reduction effects, since temperature, relative humidity, precipitation, wind and irradiance was classified to be similar to those in March 2020.

To further analyse the weather regimes for the first half of 2020 the classification proposed by Hess and Brezowsky (1977) has been chosen (see also Bissolli and Dittmann (2001)). This classification identifies predominant synoptic regimes over Central Europe and defines 30 so called `Großwetterlagen´ (GWLs), which can be isolated by an objective method introduced by James (2007). The underlying data for this analysis were provided by the German Weather Service. The results of the GWL-classification can be found in supplemented material, Table A2

### Pre-lockdown period

In February 2020, an unusually wet period occurred due to strong cyclonic activity in Central Europe. Westerly and North Westerly cyclonic regimes were observed on 76% of the days, whereas high pressure-type regimes were observed on only 24% of the days Thus, the shortwave downwelling irradiance in February 2020 is one of the lowest measured at the weather station Wettermast Hamburg (53°31' 09"N and 10°06'10"E) (https://wettermast.uni-hamburg.de) (Brümmer and Schultze, 2015) during the last 25 years (Figure A4), being representative for north western Europe. The accumulated precipitation for February at this weather station with an amount of more than 120 mm was exceptionally high compared to the last decades (Figure A4).

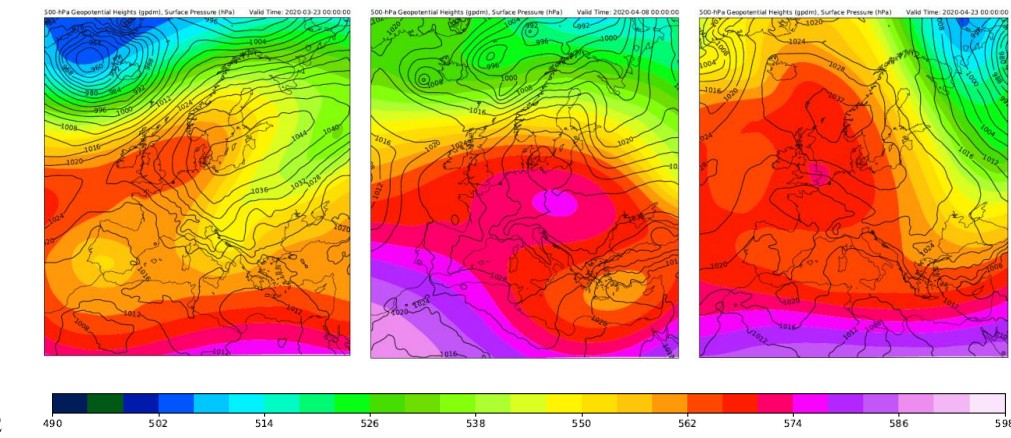





**Figure 6: 500 hPa geopotential heights (in gpdm) and surface pressure (in hPa) for selected time segments in March**
**and April 2020 according to the COSMO simulations. The geopotential heights are averaged over 4 days (21.03.-24.03;**
**6.04.-9.04., 21.04.-24.04. from left to right, respectively). Displayed surface pressure distributions are representative**
**snap shots within those time segments.**

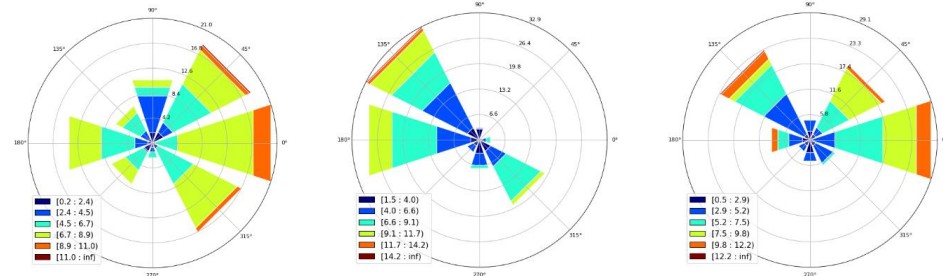


**Figure 7: Wind roses derived from measurements of the weather station Wettermast Hamburg at an altitude of 110 m.**
**Results for 3 periods covering about 15 days each are shown: 16.03. – 31.03.2020; 1.04.-15.04.2020; 16.04-30.04.2020,**
**from left to right.**

**Main lockdown period**
For the meteorological characterisation of the main lockdown period between mid of March and end of April we
rely in addition to the GWL analysis on maps of the 500 hPa geopotential height and the surface pressure
distribution. The underlying data were extracted from simulations with the COSMO-MERRA system, the same
meteorological fields, which have been used for the chemistry transport calculations with CMAQ displayed and
discussed in the following chapters. In Figure 6 a subset of those maps for 3 selected time periods is shown; the
complete set of maps generated can be found in the appendix (Fig A5). To characterise and quantify horizontal
advection, wind roses derived from observations at the Wettermast Hamburg are displayed in Figure 7. The wind
data in each plot cover a time period of about 15 days. Measurements at an altitude of 110 m were chosen to better
represent a larger area and eliminate parts of the surface influences on the wind.
In mid of March, the synoptic regime substantially changed over Europe. 'High pressure'-type GWLs became
dominant, i.e. high ridges over Central Europe and high-pressure systems led to a typical atmospheric blocking of
cyclones. The weather situation shows first a varying blocking in North- and Central Europe followed by a high
pressure ridge reaching form the Azores to Scandinavia (Figure 6, left), which changed to a high pressure ridge
stretching from Iceland into Russia. In northern Germany the wind regime was dominated by a flow with mainly
easterly components, which were relatively high wind speeds (Figure 7, left). In southern Europe the situation,
which was similar at the beginning of the period to that one in the North, changes starting about on the 23rd of
March, an isolated trough formed leading to low pressure system activity. For March 28 and 29 dust transport
from Asia and Northern Africa to the Po Valley was reported (Collivignarelli et al., 2020).
In the first half of April the weather in the north-eastern part of Central Europe was again quite variable, and in
Southern Europe the cut-off from the northern regime could still be recognized. In the western part of Central
Europe a ridge has established, which stretched towards the UK. Accordingly, winds in Northern Germany blew



predominantly from westerly/north westerly directions. Later on, a ridge over entire Central Europe dominated
the weather in the study domain (Figure 6, middle), only the Eastern Mediterranean was still influenced by a cut-
off trough. In the Po valley according to measurements around Milan, the weather during the second half of March
to April 10th was dry and very sunny with low to medium wind speeds (Collivignarelli et al., 2020). Towards the
mid of April a high pressure bridge was established reaching from Iceland into Eastern Europe.
In the second half of April a high pressure system established over the British Isles attached to a ridge located
over Central Europe leading to dry and sunny weather all over Europe. This condition was basically stable until
April 25th, when a cyclonic flow took over, leading to more westerly winds over Central Europe, a situation which
lasted until the first days of May. Winds in northern Germany switched over from easterly to more westerly
directions this time (Figure 7, right).
Overall, an exceptionally dry period occurred which started in the early lockdown period and continued until the
end of April. The weather was characterized by very low cloud cover and record-breaking large amounts of solar
irradiance (see the record at the Wettermast Hamburg in Fig. A4) and little precipitation. This exceptional weather
period is also discussed by van Heerwaarden et al. (2021), who reported record breaking solar irradiation for the
Netherlands.

**Lockdown transition**

In May 2020, atmospheric conditions were very different in Central Europe compared to the previous months. For
instance, Germany was dominated by large amounts of rain in the south, sunny conditions in the west and dry but
cloudy conditions in the east and north. Observed sunshine duration and solar irradiance corresponds
approximately to average climatic conditions. In contrast, large parts of Western Europe (Netherlands, Belgium,
West Germany, UK) experienced sunny and dry weather throughout the entire May (van Heerwaarden et al.,
2021). Finally, the large scale conditions in June turned out to favour long-lasting periods with dry and sunny
weather conditions in Northern Germany due to blocking conditions caused by high pressure systems located over
Scandinavia. However, the more southerly regions were rather too wet in a climatological sense.

**4.2 Concentrations of $NO_2$, $O_3$ and $PM_{2.5}$**

The reduced emissions of pollutants during the lockdown periods, which are pronounced in certain sectors, should
lead to changes in ambient concentrations of those substances and related secondary pollutants as ozone. Beside
regional emissions also advected pollutants and the meteorological conditions determine local and regional
concentrations. To assess changes in air quality and alterations in the behaviour and nature of concentration time
series observations at selected air quality measurement stations have been examined. The analysed stations have
been selected in a way that they are geographically distributed over the study domain and represent different
emission characteristics. The stations Radhuset in Malmö, Sweden, and Sternschanze in Hamburg, Germany, are
classified as urban background stations, not directly influenced by traffic. In Malmö, the station is located in the
historical part of the town near the town hall, the Hamburg station is placed in a park of a quite lively quarter of
the town. Both urban background stations may be influenced by ship traffic. Waldhof is a rural background station
in northern Germany located about 60 km north of the city of Hannover. Vredepeel is a background station in a
fairly populated part of the Netherlands situated in the triangle between the cities Nijmegen, Eindhoven and Venlo.
The observatory Kosetice in the Czech Republic is located in the Moravian Highlands in an agricultural





countryside about 80 km from south-east of Prague. To represent a region south of the Alps the Italian station San
Rocco in Po-Valley about 30km east of Parma has been selected. With the exception of Kosetice, having an
elevation of about 530m, the stations are situated below an altitude of 80m. To allow a comparison of the
concentration measurements under different meteorological influences time series of $NO_2$, $O_3$, and $PM_{2.5}$ for the
years 2015 to 2020 have been examined. However, $PM_{2.5}$ was not available at the station San Rocco.

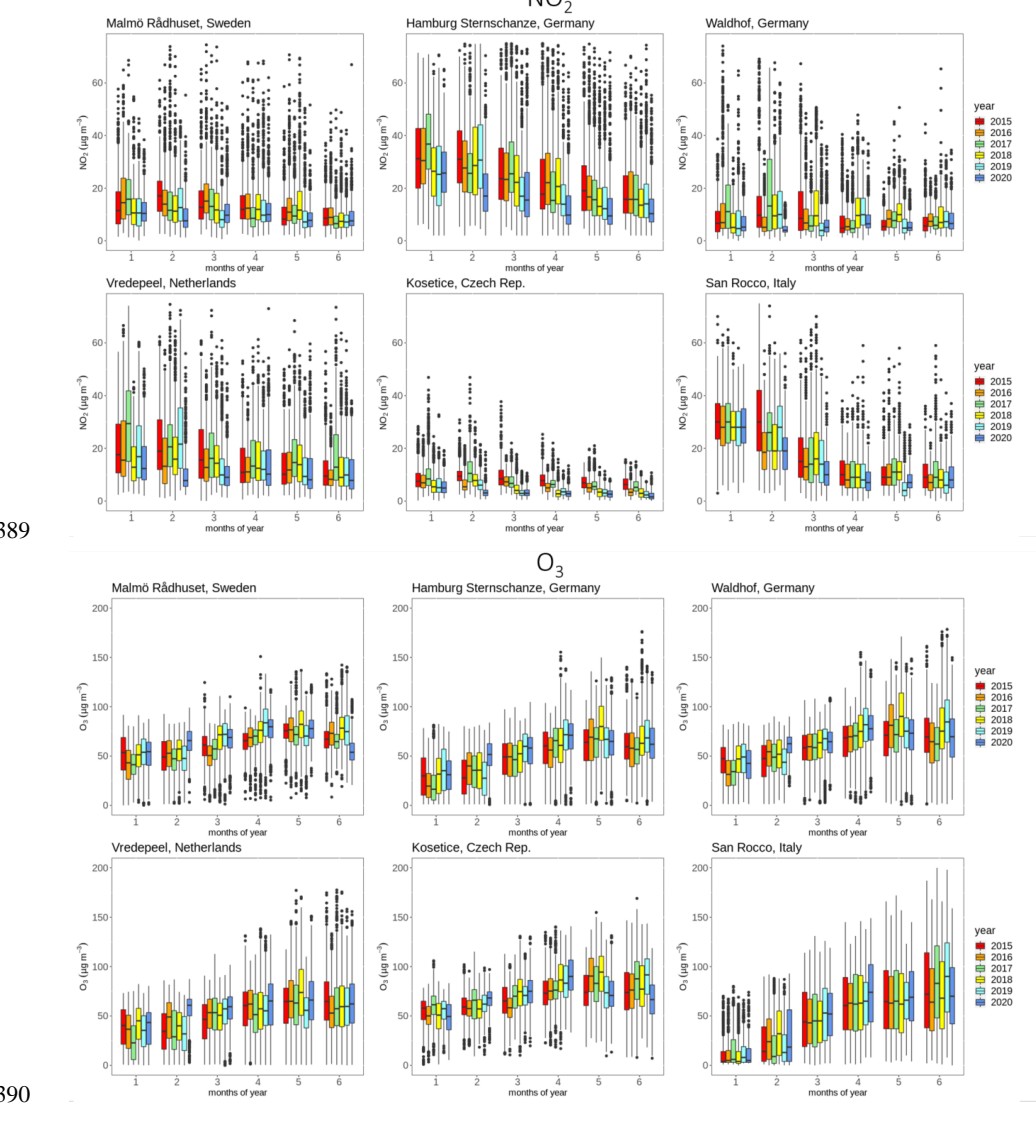



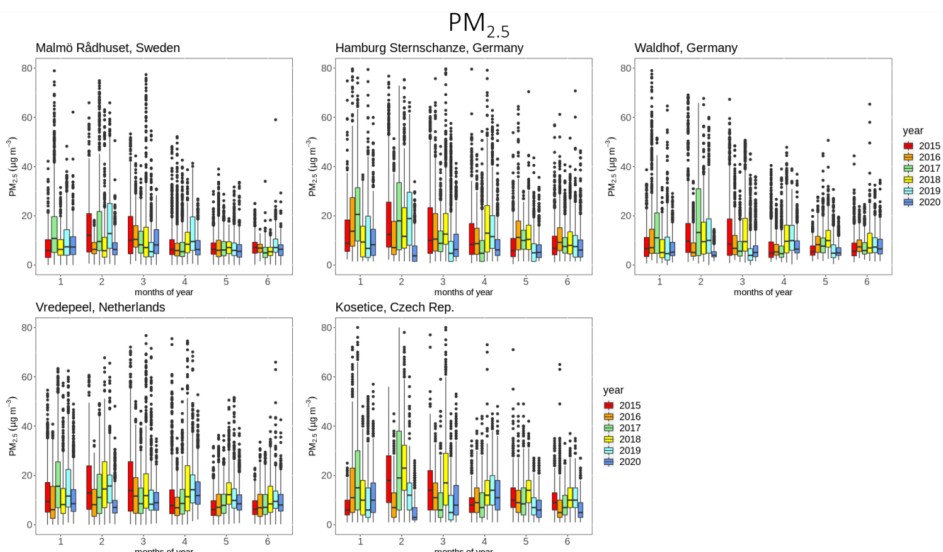

**Fig. 8: Observed monthly concentrations of NO₂, *O₃*, and PM₂.₅ at Waldhof (Germany), Vredepeel (The Netherlands), San Rocco (Italy), Kosetice (Czech Republic), Malmö (Sweden) and Hamburg (Germany). The median is displayed within the central boxes which span from the 25th percentile to the 75th percentile, called the interquartile range of the underlying frequency distributions. For NO₂ and PM₂.₅ these distributions are based on hourly measurements at the different stations and for *O₃* on daily 8 hour maximum values. The whiskers above and below the central boxes indicate the largest and the smallest value within 1.5 times the interquartile range, respectively. Dots denote values outside these ranges. PM₂.₅ was not available at San Rocco.**

The observational results for the selected stations for $NO_2$, $O_3$, and $PM_{2.5}$ are displayed in Fig. 8. For $NO_2$, at all stations, with the exception of Waldhof, an obvious trend from higher concentrations in the winter months to lower ones in spring in early summer can be seen. At Waldhof this trend is not that clear due to lower values in January for most of the years. As it can be expected, in urban (Malmö and Hamburg) or densely populated (Vredepeel and San Rocco) regions the $NO_2$ concentration are on a higher level. At most stations the $NO_2$ concentrations for March 2020, the month during which in all countries the lockdown measures started, are among the lowest ones compared to the previous years. For Hamburg, Vredepeel and Kosetice this also holds for the months April to June. An obvious feature, which appears at all stations except San Rocco is, that the February concentrations in 2020 are lower compared to the previous years, although no lockdown measures were taken in Europe in February. Presumably, meteorological conditions are responsible for these relatively low $NO_2$ concentrations. February 2020 was a month with steady westerly winds and longer periods of intense precipitation in Northern Europe. While strong winds cause rapid dilution of pollutants, steady precipitation has a cleaning effect due dissolution of pollutants in cloud and rainwater and subsequent wash-out.

For $O_3$, at all stations and for all years the typical trend from low winter concentrations to higher concentrations in spring and early summer can be seen. During the lockdown month April the $O_3$ concentrations for the years 2018, 2019, 2020 were higher than in the previous years. During those years the radiation was rather intense in April, which favours the photochemical formation of ozone. At the rural stations Waldhof and Kosetice ozone concentrations in May and June 2020 were lower than in previous years. At the urban stations in Malmö and Hamburg the relative increase in $O_3$ concentrations over the 6 month period is lower compared to the more rural stations. This can be interpreted as a titration effect of $O_3$ by reactions with NO, which has significant sources in



urban areas. In general, the observations of $O_3$ maxima do not provide any indication of significant effects related
to lockdown emission changes in 2020. Possible effects of NO emission drops in March and April 2020 might be
low and masked by meteorological conditions.
$PM_{2.5}$ concentrations also show no clear signal that would allow to relate concentrations to lockdown emission
reductions. Slightly higher concentrations and variability can be observed in winter compared to summer at all
stations. This can be related to the fact that very high PM concentrations appear in winter, only, when emissions
are high and atmospheric mixing is suppressed, e.g. during high pressure situations with advection of cold air.
Similar to the $NO_2$ concentrations, rainy and windy weather in February 2020 leads to low $PM_{2.5}$ concentrations
at all stations
**5 COVID-19 lockdown effects**
Effects of the lockdown measures on emissions were discussed in section 3. Now, CMAQ model results are
evaluated for the COV and the noCOV case during the lockdown phase. Meteorological impacts are discussed
through comparisons of CMAQ model results that were derived with meteorological data for the years 2016 and

432    2018.

**5.1 CMAQ results for Central Europe**
Differences between the CMAQ results for 2020 for the COV and the noCOV case reveal the impact of the
lockdown emission reductions on air pollutant concentrations. The magnitude of the concentration changes varies
considerably in time and space. Here, we focus our evaluation on the period with the highest emissions reductions
between 16 and 31 March 2020. During this time the most widely spread and temporally stable emission
reductions took place in Europe. Differences among weekdays and weekends and, to a limited extent, also among
different weather situations are averaged out by investigating a half-month-period. However, changing effects
over time are also discussed.
**$NO_2$ concentrations**
Figure 9 shows maps of the modelled average $NO_2$ concentrations in Central Europe between 16 and 31 March
for the case without lockdown measures (noCOV) together with the absolute and relative concentration reductions
caused by the lockdown. The $NO_2$ concentrations for the noCOV case in central Europe show the typical pattern
with highest concentrations in densely populated areas like England, Belgium, The Netherlands and western
Germany as well as northern Italy (Fig 9a). Average concentrations range between 5 and 10 µg/m³. Reductions in
$NO_2$ concentrations caused by the lockdown are highest in the same regions, also reaching several µg/m³. Relative
reductions are highest in France, Belgium, Italy, and Austria, reaching more than 40% on average. Germany, the
Netherlands, UK, southern Sweden and the Czech Republic show lower reductions between 15% and 30%. In the
following weeks, $NO_2$ concentrations stayed more or less on the same level in most parts of Europe, but the
lockdown effects decreased slightly as it could be expected from the emission changes. Overall, relative
concentration reductions were most significant in England, France, Belgium and Italy, as it was seen for the second
half of March. Maps for relative reductions due to the lockdown for six half-month periods between 1 March 2020
and 31 May 2020 are given in the appendix (Fig A6).



**Figure 9: CMAQ results for NO₂ concentrations in Central Europe between 16 and 31 March 2020. Top:**
**Concentrations without lockdown measures (noCOV run). Middle: Absolute concentration reductions due to lockdown**
**measures (noCOV – COV run). Bottom: Relative concentration reductions due to lockdown measures (noCOV – COV**
**run); positive values for absolute and relative differences denote high reductions.**



**O₃ concentrations**
It can be expected that reduced NOx emissions are also reflected in modified $O_3$ concentrations with lower values
in all regions that are NOx-limited. However, for the second half of March increased $O_3$ concentrations between
1 and 8 µg/m³ were modelled in the COV case for northern Central Europe and the Po valley (see Fig. 10). Because
these are the regions with the highest NOx emissions in Europe, they were most likely VOC-limited during this
first lockdown period and $O_3$ titration with NO was reduced when NOx emissions were reduced. Most of the
southern parts of the modelling domain exhibited a decrease in ozone of 1-2 µg/m³ on average caused by the
lockdown and the reduced NOx emissions. In the following weeks, areas with increased ozone turned smaller
week by week and were limited to large cities and the most densely populated areas, see Fig 11 for the first half
of April and the first half of May. Most regions in Europe turned into NOx-limited areas in spring 2020, resulting
in lower ozone concentrations of 1-2 µg/m³ (about 2-4% change) caused by the emission changes during the
lockdown (see Fig.A7 in the supplement).

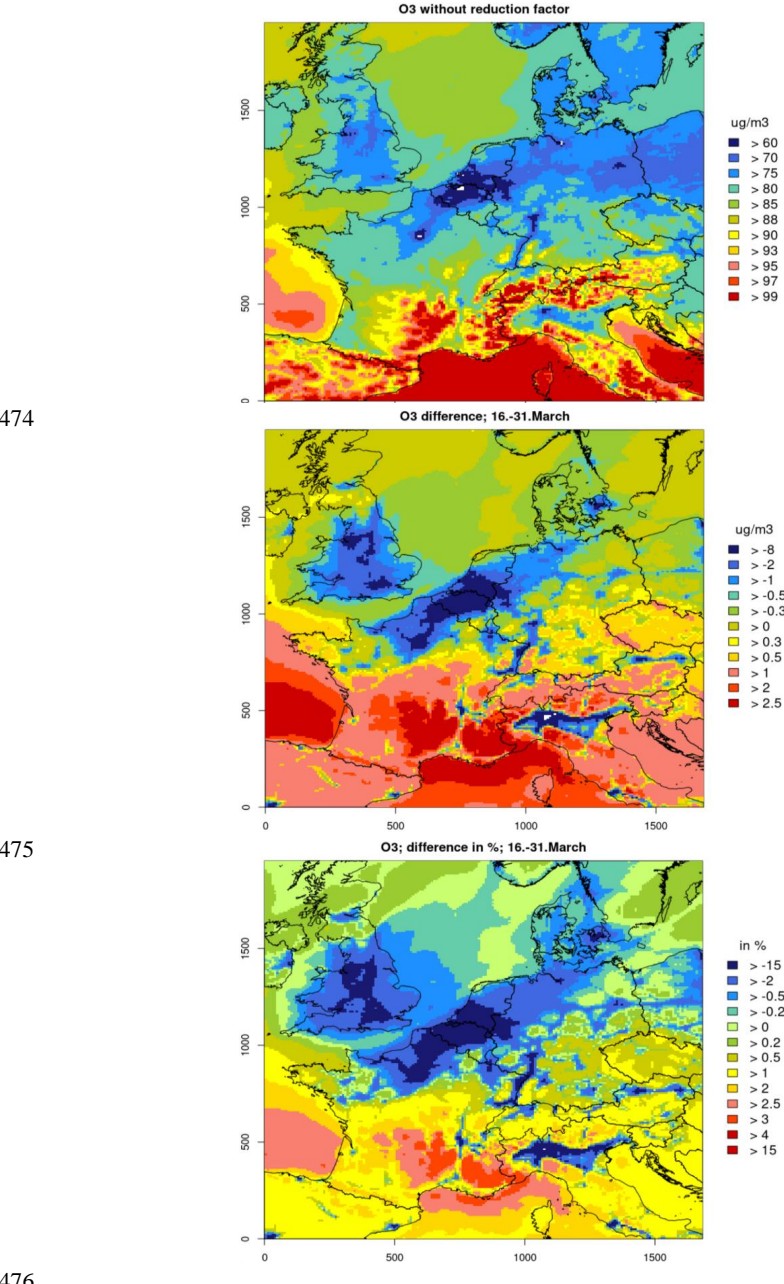





**Fig. 10: CMAQ results for O₃ concentrations in Central Europe between 16 and 31 March 2020. Top: Concentrations**
**without lockdown measures (noCOV run). Middle: Absolute concentration reductions due to lockdown measures**
**(noCOV – COV run); positive values denote high reductions. Bottom: Relative concentration reductions due to**
**lockdown measures (noCOV – COV run); positive values denote reductions, negative values denote increases.**






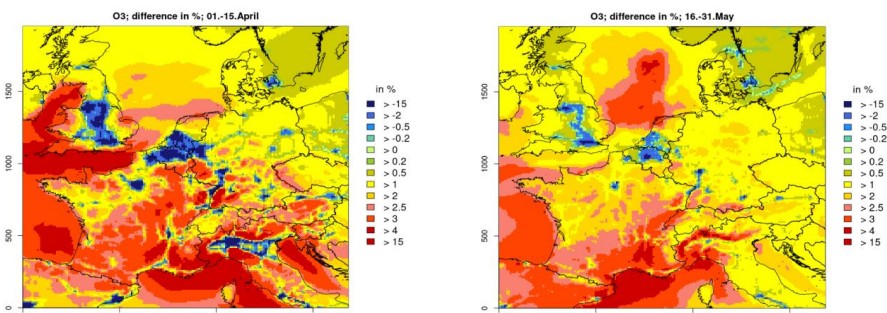


**Fig. 11: CMAQ results for changes in O₃ concentrations due to lockdown measures in Central Europe between 1 and 15 April 2020 (left) and 16-31 May 2020 (right). Positive values denote concentration reductions, negative values denote concentration increases.**

**PM2.5 concentrations**

Simulated $PM_{2.5}$ concentrations in the second half of March 2020 for the noCOV case show relatively high concentrations between 12 and 15 µg/m³ in large parts of Central Europe and the Po valley while the UK, Denmark and Northern Germany exhibited concentrations below 10 µg/m³ (see Fig. 12, top). The lockdown emission reductions lead to concentration reductions between 1 and 3 µg/m³ in those regions with higher concentrations and values below 1 µg/m³ in the north western part of the domain. Relative concentration decreases were most significant in France and Northern Italy with values up to 20% while in the rest of the domain 6-10% lower $PM_{2.5}$ was simulated. In the following weeks, $PM_{2.5}$ concentrations were typically reduced by 10-20% because of the lockdown measures in most parts of Central Europe. Somewhat lower values were found in the Northern and southern parts of the domain. The reduction in $PM_{2.5}$ concentrations decreased to 6-12% in the second half of May (see Fig. A8 in the supplement).







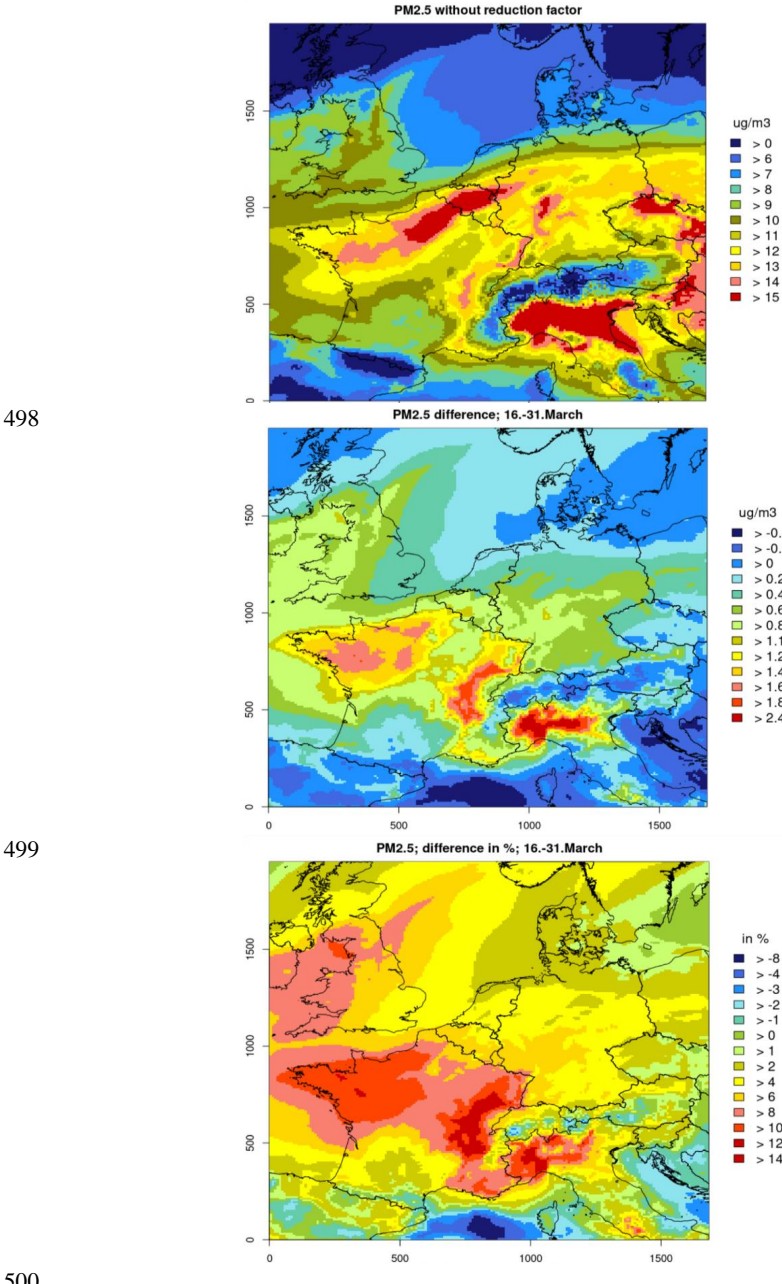

**Fig. 12: CMAQ results for PM2.5 concentrations in Central Europe between 16 and 31 March 2020. Top:**
**Concentrations without lockdown measures (noCOV run).Middle: Absolute concentration reductions due to lockdown**
**measures (noCOV – COV run); positive values denote reductions. Bottom: Relative concentration reductions due to**
**lockdown measures (noCOV – COV run); positive values denote reductions.**





**Temporal development of concentration changes**

The detailed temporal development of the effect of lockdown emission reductions on atmospheric concentrations of $NO_2$, $O_3$ and $PM_{2.5}$ is followed at selected measurement stations. Figure 13 shows the modeled differences between the noCOV and the COV model runs at Waldhof, Vredepeel, and San Rocco. Lockdown emission reductions lead to reduced concentrations of $NO_2$ and $PM_{2.5}$ at all stations, however, the amount varies considerably in time and by station. At Waldhof, only very small changes are observed. At Vredepeel, $NO_2$ is significantly reduced (by more than 10 µg/m³ on individual days) $PM_{2.5}$ shows only small reductions. At San Rocco, both, $NO_2$ and $PM_{2.5}$ are reduced by several µg/m³ until the end of April. In May and June, lockdown effects on the concentrations get much smaller, also at Vredepeel and San Rocco.

$O_3$ shows higher values despite the emission reductions until mid of April at Vredepeel and San Rocco. This is because these stations are in VOC-limited areas at that time, where NOx emission reductions lead to decreased $O_3$ titration. This pattern changes towards end of April and in the following $O_3$ is decreased on most of the days at all stations as a consequence of lower NOx emissions. This effect remains variable at Vredepeel, a station close to the region with highest NOx emissions in Europe. At Waldhof, $O_3$ reductions are observed between beginning of April and end of June. On average between 16 March and 30 June, $O_3$ is only decreased by 0.6 µg/m³ (< 1%) at Vredepeel. At Waldhof and San Rocco, the reductions are 1.2 µg/m³ (1.6%) and 1.5 µg/m³ (1.9%), respectively.

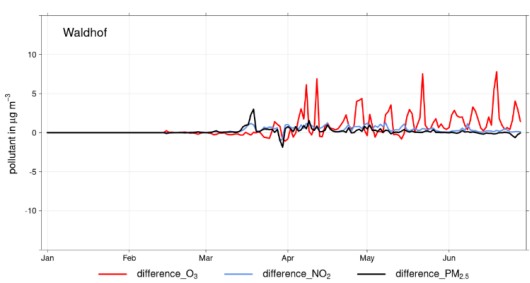

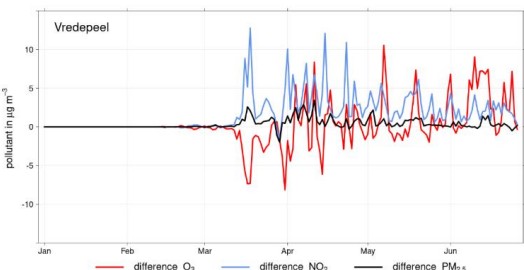



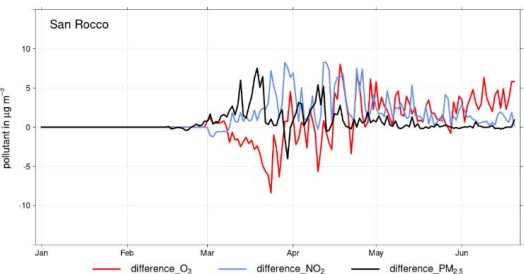


**Fig. 13: Temporal development of the differences in the simulated concentrations of $O_3$ (red), $NO_2$ (blue) and $PM_{2.5}$ (black) in Waldhof (top), Vredepeel (middle) and San Rocco (bottom) between 1 January and 30 June 2020.**

**5.2 Impact of meteorological conditions**
For investigating the effects of the exceptional meteorological situation on the concentration reductions in March
and April 2020, additional CMAQ model simulations were performed. Meteorological data simulated with
COSMO-CLM for the first six months in 2016 and 2018 was used as input data, together with the 2020 emissions
for both, the COV and the noCOV case. Biogenic emissions were also kept the same for the 2016 and 2018 runs
in order to investigate effects of meteorological conditions, only. These additional years were selected to cover a
span of weather situations during the lockdown phase. The selected years were different, but represent not in any
sense an extreme situation. They were chosen from the time span 2015 to 2019, since for these years model data
generated using the same advanced model settings (model version and reanalysis data) is available. The results
show the concentration and the changes caused by the lockdown measures as they would have happened under
different meteorological conditions.
Fig.14, top, shows the $NO_2$ concentration changes for 2020 relative to 2018 and 2016 caused by meteorological
conditions, only, for the period between 16 March and 30 April. No emission changes because of the lockdown
were assumed for this investigation. Meteorological conditions in 2020 caused between 20% and more than 30%
lower $NO_2$ concentrations in large areas of the North Eastern model domain (The Netherlands, Northern Germany,
Denmark and Southern Sweden) compared to 2018, even without any lockdown measures. On the other hand, in
western UK, Belgium, Northern France, and the Czech Republic, meteorological conditions lead to 20% to more
than 30% higher $NO_2$ concentrations. The picture is similar when compared to 2016, in particular in the western
part of the model domain, but the area with lower $NO_2$ concentrations in 2020 compared to 2016 does not include
the North Sea and Denmark.



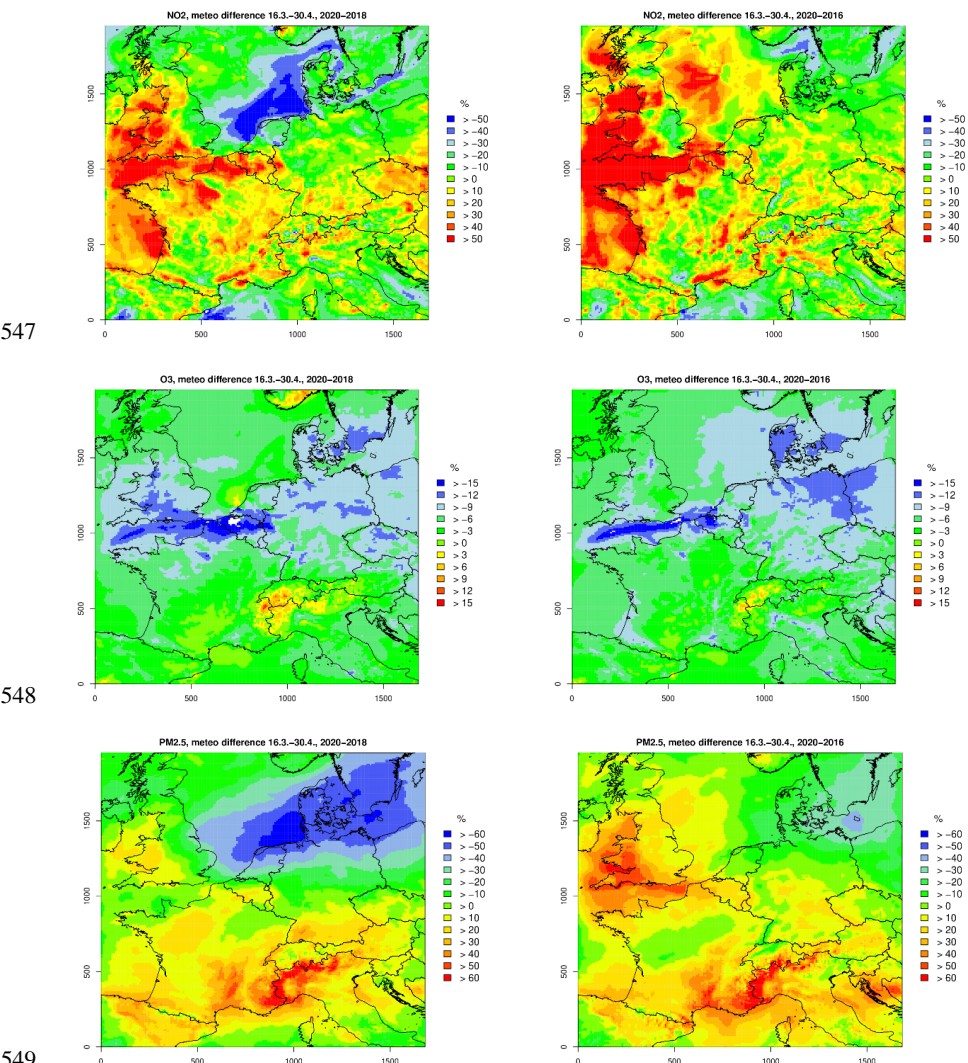

**Fig. 14: Relative concentration changes due to meteorological conditions in Central Europe between 16 March and 30 April simulated with CMAQ for $NO_2$ (top), $O_3$ (middle) and $PM_{2.5}$ (bottom): The changes are represented as relative numbers for 2020 compared to 2018 (left) and 2016 (right). Positive values denote higher concentrations in 2020 relative to the previous year. Be aware of the different scales for each pollutant.**

Average ozone concentrations between 16 March and 30 April 2020 were relatively low in almost entire Central Europe when compared to a situation with meteorological conditions as in 2018 and 2016 (see Fig. 12, middle). Differences are in the order of 10-15% in the northern part of the model domain and between 2 and 6 % in the southern part. Only in few spots in Northern Italy and Southern Switzerland, the meteorological situation in 2020 favoured ozone formation compared to 2016 and 2018.

The picture is more mixed for $PM_{2.5}$ with considerably lower concentrations in 2020 compared to 2016 and 2018, particularly in Northern Germany and Poland, i.e. in the north eastern part of the domain. Relative differences reach more than 50% between 2020 and 2018 in the German Bight. Compared to 2018, $PM_{2.5}$ concentrations were

also low in the western UK in 2020. In almost entire France and in Northern Italy, PM$_{2.5}$ concentrations were
relatively high in 2020 compared to 2016 and 2018, differences again reach more than 50% but with opposite
sign.



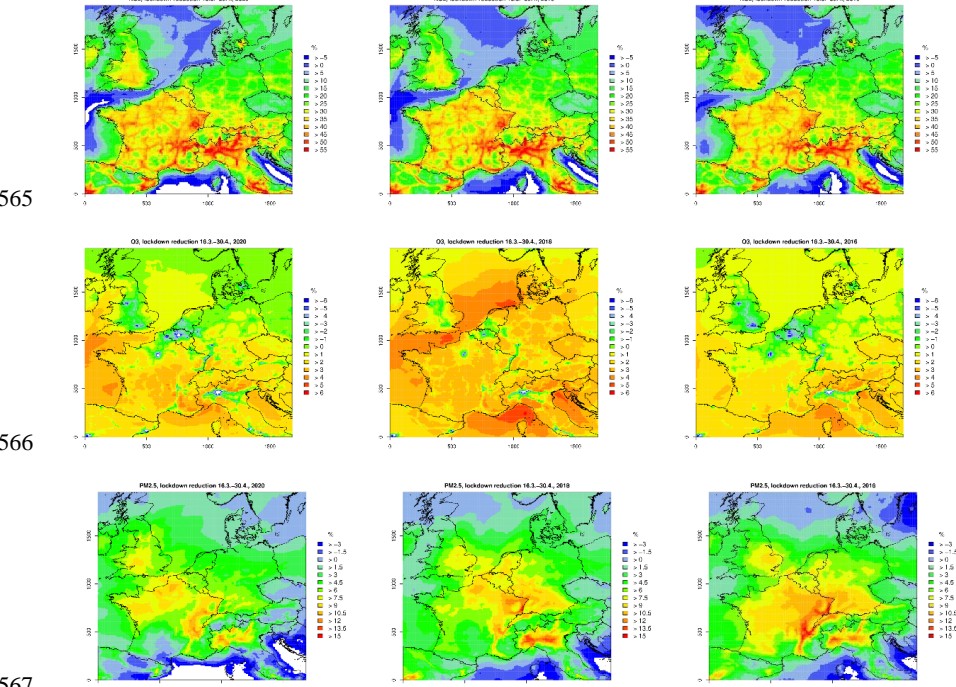

**Fig. 15: Relative concentration reductions due to lockdown measures (noCOV – COV run) in Central Europe between**
**16 March and 30 April simulated with CMAQ for NO$_2$ (top), O$_3$ (middle) and PM$_{2.5}$ (bottom) and three different**
**meteorological input data sets. Left: 2020, Middle: 2018, Right: 2016. Positive values denote concentration reductions**
**caused by the lockdown emission changes. Be aware of the different scales for each pollutant.**
The meteorological situation also affects the concentration changes caused by the lockdown, but this differs
considerably among the pollutants. Fig 15 shows the lockdown emission reduction effects on the average
concentrations for the main lockdown period from 16 March to 30 April. In most parts of Central Europe the
variation for NO$_2$ is rather small (plus/minus approx. 5%). For ozone, on the other hand, effects of the lockdown
are quite different among the three selected meteorological years. For 2020 meteorological conditions, relatively
large areas in Northern Central Europe show a slight increase in ozone (green and blue areas in Fig. 15, middle
row). These areas would have been smaller with 2016 meteorological conditions and limited to the most densely
populated areas for 2018 meteorological conditions. Lockdown effects on PM$_{2.5}$ would have been more significant
under meteorological conditions of the years 2016 and 2018 in almost the entire model domain (Fig. 15, bottom
row). Particularly in Northern Italy and South East France, changes in PM$_{2.5}$ caused by the lockdown could be
more than 10%, a value that was rarely reached during the real lockdown in 2020.



**6 Discussion**
**6.1 Time series at selected stations**
Observations of $NO_2$ and $PM_{2.5}$ concentrations in Central Europe in the first six months of 2020 showed low
concentrations in March and April when compared to previous years. According to CMAQ model simulations
that consider lockdown emission reductions as well as emissions that could be expected for 2020, the lockdown
effects are strongest for $NO_2$ with average concentration reductions up to 40% between mid of March and mid of
April. $PM_{2.5}$ shows reduction up to 20% while the effect on $O_3$ is much lower (up to 4% reduction). $O_3$
concentrations might even increase in large parts of northern Europe in March.
In order to quantify the quality of these model estimates, the simulated concentrations were compared to
observations at selected stations (including those presented in section 4 and 5). Figure 16 exemplarily shows the
comparison at Vredepeel, Table 1 contains statistical values for $NO_2$ and $O_3$ at 11 stations and for $PM_{2.5}$ at 4
stations in Europe.
Modelled $NO_2$ concentrations are typically lower than the observed values, in particular, the model shows a
stronger downward trend of the concentrations in spring than observed. This pattern is reversed for ozone, where
the modelled 8h max concentrations are typically too high with better agreement in spring compared to winter.
$PM_{2.5}$ is underestimated on average, but only at 2 out of 4 stations. Here, the agreement is typically better in winter
compared to spring. As average for all selected stations, the model bias for $NO_2$ is -17%, for $O_3$ it is +21% and
for $PM_{2.5}$ it is -5%. The temporal correlation ($R^2$) based on daily mean values varies between 0.42 and 0.74 for
$NO_2$, between 0.07 and 0.75 for $O_3$ and between 0.21 and 0.62 for $PM_{2.5}$. Details are given in Table 1.
The model is able to reproduce observed concentration levels and their spatiotemporal variation. The agreement
between modelled and observed concentrations is in a range that tis typical for regional CTMs (see e.g. Solazzo
et al. (2012)). The deviations from the observed values can be interpreted as relative uncertainties in the modelled
lockdown effects. During the lockdown between March and June, deviations between modelled and observed
concentrations are often higher than the changes caused by the lockdown. Therefore, the results cannot be used to
judge how accurate the estimated emission reductions are.
Based on the 6 months of simulation, the average concentrations reductions at the 11 selected stations are 14%
(7%-26%) for $NO_2$, 0.4% for $O_3$ (-1.3% to +1.7%) and 2.3% for $PM_{2.5}$ (0.1% to 4.0%). While half year average
$NO_2$ concentrations in highly polluted areas decreased between 15% (Vredepeel, The Netherlands) and 25%
(Besenzone and Casirate d'adda, Po Valley, Italy), $NO_2$ reductions are much smaller (7-15%) in rural areas.
Average $O_3$ concentrations increased slightly (1%) close to cities and decreased in rural areas (up to 2%). For
$PM_{2.5}$, concentration changes at the four measurement stations were mostly between 2 and 4%. Under the
assumption that emission reductions were much lower in the second half of 2020, the lockdown emission
reductions exhibit only very small effects on annual average pollutant concentrations, especially for secondary
pollutants. Concentration reductions at the measurement stations for the main lockdown period (16 March – 30
April) are also given in Table 1.






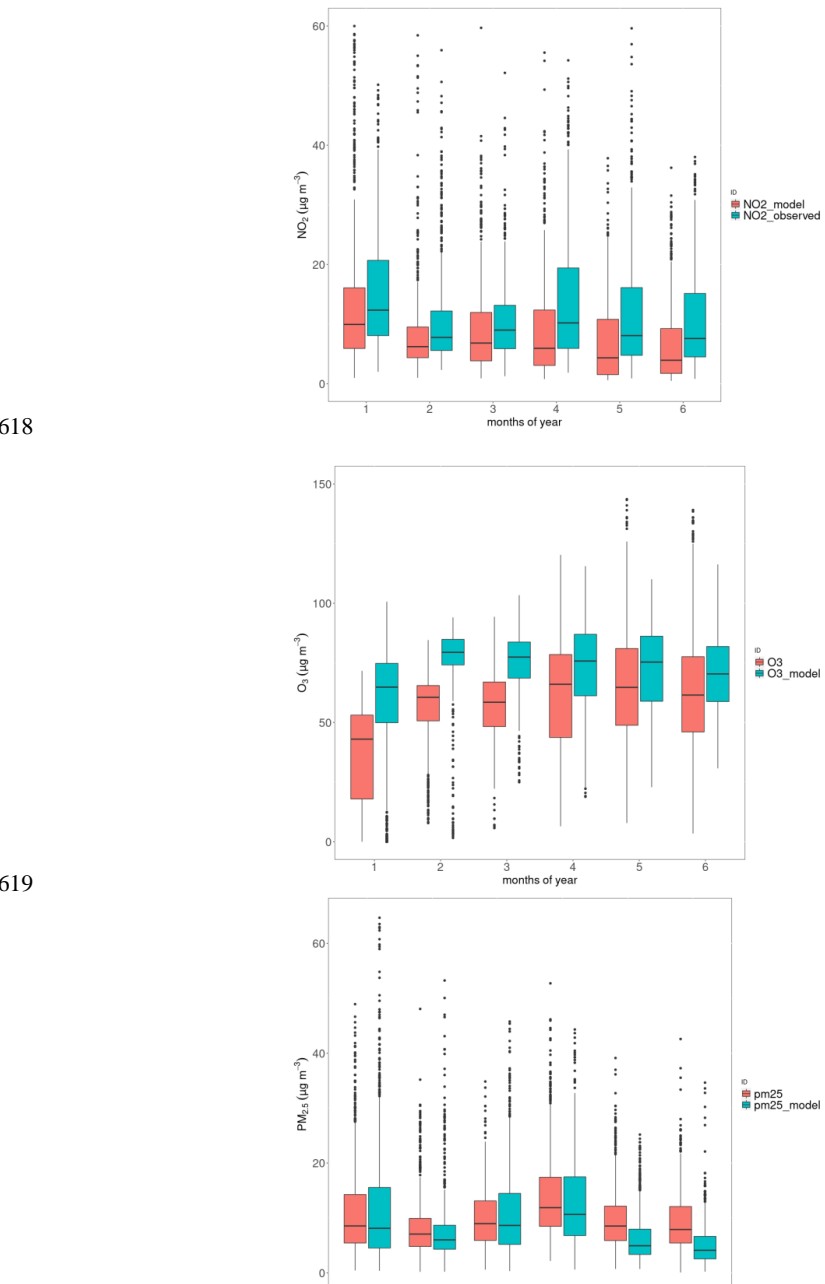

**Fig. 16: Comparison between model results (green) and observations (red) at Vredepeel, Netherlands. Top: NO₂,**
**middle: O₃, bottom: PM₂.₅. All concentrations are given in µg/m³, box plots show medians, 25% and 75% quartiles and**
**whiskers representing 1.5 times the interquartile range. Values that fall outside the range of the whiskers are given as**
**dots.**





**Table 1: Statistical evaluation of a comparison between observations of NO₂ at selected background stations of the EEA**
**network with CMAQ model results between 1 Jan 2020 and 30 June 2020**

| NO₂ concentrations 1 Jan 2020 – 30 June 2020 | | | | | |
|---|---|---|---|---|---|
| Station | Observed [µg/m³] | Modelled (COV case) [µg/m³] | Bias (model-obs) [µg/m³] | Correlation | Lockdown effect COV-noCOV (16.3.– 30.4.) [µg/m³] |
| Risoe, DK | 4.7 | 5.7 | 1.0 | 0.46 | -3.0 |
| Waldhof, DE | 5.0 | 3.8 | -1.2 | 0.63 | -0.6 |
| Zingst, DE | 4.4 | 2.9 | -1.5 | 0.63 | -0.4 |
| Neuglobsow, DE | 2.9 | 2.6 | -0.3 | 0.66 | -0.5 |
| Vredepeel, NL | 12.4 | 10.2 | -2.2 | 0.64 | -3.7 |
| De Zilk, NL | 11.4 | 12.8 | 1.4 | 0.51 | -3.7 |
| Kosetice, CZ | 3.4 | 3.0 | -0.3 | 0.42 | -0.6 |
| San Rocco, IT | 13.5 | 9.2 | -4.3 | 0.74 | -3.7 |
| Besenzone, IT | 15.8 | 11.9 | -3.9 | 0.71 | -7.3 |
| Casirate d'adda, IT | 19.4 | 15.9 | -3.5 | 0.71 | -10.5 |
| Paray le Fresil, FR | 3.1 | 2.1 | -1.0 | 0.54 | -0.9 |
| O₃ concentrations 1 Jan 2020 – 30 June 2020 | | | | | |
| Risoe, DK | 71.2 | 75.7 | 4.5 | 0.07 | 0.5 |
| Waldhof, DE | 63.6 | 74.5 | 10.9 | 0.25 | -0.7 |
| Zingst, DE | 70.6 | 79.7 | 9.1 | 0.23 | -0.5 |
| Neuglobsow, DE | 62.8 | 74.8 | 12.0 | 0.16 | -0.6 |
| Vredepeel, NL | 56.8 | 70.5 | 13.7 | 0.55 | -0.3 |
| De Zilk, NL | 63.1 | 70.6 | 7.5 | 0.34 | 0.0 |
| Kosetice, CZ | 70.0 | 78.6 | 8.6 | 0.21 | -1.0 |





| | | | | | |
|---|---|---|---|---|---|
| San Rocco, IT | 54.7 | 73.4 | 18.7 | 0.68 | -0.9 |
| Besenzone, IT | 49.5 | 69.3 | 19.8 | 0.59 | 0.7 |
| Casirate d'adda, IT | 56.3 | 74.0 | 17.7 | 0.75 | 1.0 |
| Paray le Fresil, FR | 58.6 | 77.2 | 18.6 | 0.43 | -1.3 |
| PM$_{2.5}$ concentrations 1 Jan 2020 – 30 June 2020 | | | | | |
| Waldhof, DE | 6.8 | 7.3 | 0.5 | 0.21 | -0.1 |
| Vredepeel, NL | 10.6 | 9.2 | -1.4 | 0.57 | -0.4 |
| De Zilk, NL | 6.8 | 7.8 | 1.0 | 0.44 | -0.2 |
| Kosetice, CZ | 9.3 | 7.8 | -1.5 | 0.62 | 0.0 |


## 6.2 Emission estimates


Emissions for 2020 were estimated based on data for 2016 and extrapolation factors that resemble the temporal
development of total sectoral emissions during 3 years before 2016. This method leads to emission corrections
that are typically on the order of 10 % but may be up to 40%. This method bears some uncertainties, however in
countries that have a high share in the total emissions in Central Europe, emission trends were rather stable during
the last 20 years. Good agreement between observed and modelled concentrations during the weeks before the
lockdown gives confidence in the method.
Estimates for lockdown emission reductions also include several sources of uncertainty. Reduction of NOx
emissions from traffic have the largest share in the emission reductions. In this approach, the LAFs applied are
based on google mobility data that resembles all traffic activities, regardless of their real emissions. I.e. no
distinction between trucks and small private cars is made and it seems likely that traffic related to transporting
goods was less reduced than private and commuter traffic. Therefore, emission reductions in traffic might be
overestimated. On the other hand, possible emission increases for residential heating that are related to more
people working from home were not considered at all. Small changes in other sectors like off-road machinery that
might have taken place weren't considered, either.
The cubic spline interpolation, applied to derive daily LAFs from monthly statistical data, enables to represent the
mean of each month correctly while giving an assumption on the daily values with a rather smooth curve. This
assumption does not necessarily represent the real daily conditions as extrema in the interpolation always occur
at the start or in the middle of the month, which might not be the case in reality. However, it is an improvement
compared to using monthly averages for each day of the month, as in this case, extreme jumps can occur at the
transition to the next month that author's assume to be more unrealistic. In addition it might resemble the rapid
emission reductions mid of March better than a monthly value.



The modelled reductions in $NO_2$ concentrations close to ground which are 30-40% on average during the second
half of March are close to what was estimated from satellite observations. (Bauwens et al., 2020) report columnar
$NO_2$ reductions of approx. 20% around Hamburg, Frankfurt and Brussels, 28% for the area around Paris and 33 –
38% for Northern Italy. Such values are in quite good agreement with the modelled values in this study.
**6.3 Impact of meteorological conditions on lockdown effects**
Meteorological conditions play a major role for concentrations of air pollutants. Not only emissions, but also
atmospheric transport and chemical transformation, as well as wet and dry deposition influence atmospheric
concentrations of $NO_2$, $O_3$ and $PM_{2.5}$. To further assess the influence of meteorological conditions on
concentrations of pollutants over Europe, CMAQ was run using emission data for 2020 (noCOV case) but
combined with meteorological input data for two different years, namely 2016 and 2018. These years were
selected, because they represent significantly different meteorological conditions. In the following, the differences
to the year 2020 for the days between 16 March and 30 April, the period that is further investigated, are briefly
summarized. In the supplement (Fig. A9 - A11) relevant plots showing differences for the meteorological
parameters 500 hPa geopotential height, total precipitation and global solar radiation can be found. The results are
based on the COSMO-CLM simulations for the respective years. It should be noted that the simulations for 2016
and 2018 do not resemble the real situation during these years, because all emissions and chemical boundary
conditions were for 2020.
**Meteorological differences 2020 versus 2016 and 2018**
In 2020 the geopotential height at 500 hPa over the British Isles and the North Sea was significantly higher
compared to that in 2016, especially from 1 April onward. This resulted in a constellation, which favours blocking
in 2020. Near surface high pressure systems were amplified and more persistent and weak wind conditions and a
more continental flow dominate. In 2016 stronger winds of Atlantic origin occasionally were observed. In 2020
precipitation was considerably lower compared to 2016. In most parts of the study region solar radiation was
clearly higher in 2020, especially over Central Europe up to the British Isles.
Much of what was has been said concerning the blocking condition in 2020 holds as well when compared to 2018.
The year 2020 also was much drier and incoming solar radiation was more intense. In 2018 winds had a more
easterly to south-easterly component. The spatial and temporal distribution and the absolute values of the
meteorological parameters were slightly different in 2018 compared to 2016 (see Fig. A9- A11), so this year
became an additional choice for the evaluation of meteorological influences.
**$NO_2$ concentrations**
During the six weeks of the most stringent lockdown measures in Central Europe (16 March to 30 April), emission
reductions caused $NO_2$ concentrations reductions between 15% and more than 50%. These reductions are almost
independent of the meteorological situation, as can be seen in Fig 15 (top row). Differences in modelled $NO_2$
concentrations between 2020 and 2016 or 2018 show variations of more than 30%, but they are fluctuating in both
directions on small spatial scales (see Fig. 14, top row). Larger areas with systematic differences are mainly found
over sea and in areas with relatively low average concentrations, like in the western UK. It can be concluded that
the $NO_2$ concentration reductions during the lockdown were dominated by the emission reductions and not very





much by the meteorological situation. This is in agreement with the fact that $NO_2$ concentrations are spatially
closely connected to the emission sources. $NO_2$ is quickly formed from NO after the latter was emitted into the
atmosphere. It will then react further to form $O_3$ at daytime. Compared to $O_3$ and secondary PM, $NO_2$ is a rather
short-lived gas with high spatial gradients and a clear annual cycle. However, as the situation in February 2020
shows, very unusual meteorological conditions, can also cause large deviations from expected concentrations.
**$O_3$ concentrations**
Ozone concentrations depend more strongly on weather conditions and on emissions of other precursors like
VOCs. Therefore, meteorological variations from year to year might have a much stronger influence on average
concentrations than the emission reductions during the lockdown. The six-weeks-average ozone concentrations
vary by +/- 15% between 2020 and 2016 or 2018 (Fig 14, middle row) while the lockdown effects are mostly in
the range of +/- 5% (Fig 15, middle row), except in densely populated areas. Weather conditions between 16
March and 30 April 2020 favoured relatively lower ozone concentrations in most parts of Central Europe when
compared to 2016 and 2018. In the simulations, only areas in the western Alpine region show higher ozone in
2020 (Fig 14, middle row). First of all, this is surprising because 2020 was comparably sunny and dry, which
should favour ozone formation. However, advection of relatively clean air from Scandinavia into the North
Eastern part of the model domain led to lower ozone concentrations particularly in the second half of April. A
comparison of the meteorological effects on $NO_2$ and $O_3$ in Fig 14 also shows that $NO_2$ was relatively high and
$O_3$ relatively low in 2020 in the English Channel, in south western UK and Belgium. The high pressure situation
with relatively low wind speeds in 2020 resulted in efficient ozone destruction at night in areas with high NO
emissions.
Lockdown emission reductions caused relative ozone increases in urban areas and throughout the northern part of
the model domain, because these areas are VOC-limited regions. For northern Central Europe this is connected
with advection of clean air from north east. Lockdown effects on ozone might differ in sign under different
meteorological conditions, as can be seen in Fig 15. About 2-4% $O_3$ concentration reductions in most parts of
Central Europe could have been expected with 2018 meteorological fields, when solar radiation was lower but
more southerly winds prevailed in northern Central Europe. On the other hand, with 2016 meteorological
conditions ozone changes would show similar patterns as 2020. Ozone chemistry depends on radiation,
precipitation, atmospheric mixing and the availability of precursors in a complex way. The response of ozone
concentrations to emission changes is therefore not straightforward to predict.
**$PM_{2.5}$ concentrations**
$PM_{2.5}$ is another secondary pollutant that depends strongly on weather conditions, but emission reductions will
primarily lead to concentration reductions (see Figures 12 and 13). However, the strength of this effect might also
vary considerably with meteorological conditions. Fig 14 (bottom row) shows that the main lockdown period in
2020 was favourable for $PM_{2.5}$ formation in most parts of Central Europe, with often 20% to 50% higher $PM_{2.5}$
concentrations compared to other meteorological situations. An exception is the north eastern part of the model
domain, where the meteorological situation in 2020 led to much lower $PM_{2.5}$ concentrations compared to 2018
(more than 50% lower) and 2016 (20-40% lower). Similar to the situation for ozone, this is connected to the
easterly and north easterly winds and the advection of clean air. Consequently, lockdown emission reductions had



only very minor effects on $PM_{2.5}$ concentrations in 2020 in southern Sweden, Denmark, Poland and northern
Germany. Higher $PM_{2.5}$ reductions would have been observed in most parts of Europe with 2016 and 2018
meteorological conditions. This can be interpreted in a way that the main lockdown period in 2020 was favourable
for $PM_{2.5}$ formation in large parts of Europe leading to smaller relative $PM_{2.5}$ concentration reductions, given that
the emission changes are the same.
Summarized, it can be said that the effects of lockdown emission reductions depend strongly on the meteorological
situation and that concentration changes because of weather conditions might be stronger than those of large
emission changes during a six weeks period in spring. However, this mainly holds for the secondary pollutants $O_3$
and $PM_{2.5}$, while the effects on $NO_2$ concentrations are less pronounced. Particularly changes in $O_3$ concentrations
are difficult to predict because of the complex emission-chemistry-meteorology interactions.
**7 Conclusions**
In this study, emission reductions during the first and most significant lockdown phase in Europe are estimated
from available mobility data, AIS ship position data and statistical data about industrial production and energy
use. They are applied to European emission data that is updated for 2020 following recent emission trends in
individual countries and sectors. Through meteorological and chemistry transport modelling with the COSMO-
CLM/CMAQ model system for Europe, and in higher spatial resolution for Central Europe, lockdown effects on
air pollutant concentrations are calculated. These are put into perspective with available observational data and
with modelled concentration changes from year to year that can be caused by varying meteorological conditions
for the same time of the year. The following conclusions can be drawn from this investigation.
Lockdown emission reductions in spring 2020 in Central Europe are significant, in particular those in traffic.
Other sectors, like shipping, might be of regional importance, but emission changes for this sector are less certain.
Aviation shows the largest relative reduction among the emission sectors considered, however the contribution to
the total emissions reductions is small because of its low share in total NOx emissions. Consequently, strongest
lockdown emissions reductions are seen for cities. The period with largely reduced emissions was limited to a few
weeks and emissions increased again towards mid of 2020.
In absolute numbers, concentration reductions are strongest for $NO_2$ in cities and for larger areas in the Po valley
with more than 6 µg/m³ for a two weeks average in the second half of March. Northern Italy also shows the
strongest relative decline with more than 50%. Rural areas in Germany, Poland and the Czech Republic show the
lowest reductions between 10% and 20%.
Ozone concentrations were often reduced, but not in cities and not in northern Europe between mid of March and
beginning of April. This can be explained by reduced titration in cities (NO - $O_3$ reactions that destroy ozone)
during the first phase of the lockdown, when NO emissions were lowest. However, when VOC emissions increase
in spring, most regions turn into NOx-limited areas, which means that ozone concentrations also decrease when
NOx emissions decrease. The $O_3$ concentration changes are around +/- 5% which is much less than the $NO_2$
changes. The impacts of meteorological conditions can be much larger and the temporary $O_3$ increase in north
east Europe in March would not have taken place under meteorological conditions as they were present in the
years 2016 and 2018.
$PM_{2.5}$ concentrations are also decreased because of the lockdown emissions reductions, but the magnitude is much
smaller than for $NO_2$, only between 2-10 %. Again, concentration changes can be much larger due to



meteorological conditions. The reductions in 2020 were relatively lower compared to the effects with 2016 and
2018 meteorological conditions.
Because the meteorological effects on concentrations of $O_3$ and $PM_{2.5}$ are larger than the lockdown emission
reduction effects, it is difficult to judge or even quantify emission reduction effects by observations and
comparison with previous years, only. For $NO_2$, this is different, but in exceptional situations, like in February
2020, $NO_2$ can also be strongly influenced by meteorological conditions and lead to lower concentrations than in
March during lockdown conditions.
Meteorological and chemistry transport models need to be applied to investigate the effects of emission reductions
and separate them from meteorological effects. Although these models have deficiencies and systematic errors,
e.g. underestimation of $NO_2$ and $PM_{2.5}$ concentrations, the impacts of emission changes caused by the lockdown
can be quantified. The effects in absolute numbers might be lower by the same magnitude as the model
underestimates $NO_2$ and $PM_{2.5}$. The model accuracy is not sufficient to judge the correctness of the emission
reduction estimates, however, the calculated $NO_2$ reductions agree well with estimations from ground based and
satellite observations for Central Europe.
The emission reductions for several weeks during the first COVID-19 lockdown in Europe were the largest since
decades. They can be seen as a huge test for emission reductions that could be achieved with significantly reduced
car traffic and air traffic. The reductions resulted in much lower $NO_2$ concentrations, particularly in cities, but the
effects on secondary pollutants like ozone and $PM_{2.5}$ were limited and are hard to predict. The latter holds
particularly for ozone that might even increase in some areas when traffic emissions are decreased. Year-to-year
variability caused by meteorological conditions has larger impacts on $O_3$ and $PM_{2.5}$ than the lockdown emission
changes. This implies that systematic changes in prevailing weather situations that might appear due to climate
change could mask effects of emission reductions on secondary pollutants. The relatively short duration of strong
lockdown measures also results in limited effects on annual average $NO_2$ concentrations. Depending on location,
only between 3% and 15% lower values could be reached.
**Acknowledgements**
The Community Air Quality Modeling System (CMAQ) is developed and maintained by the US EPA. Its use is
gratefully acknowledged.
We thank to the weather mast group of the Meteorological Institute at the University of Hamburg, who delivered
data from tower site Wettermast Hamburg.
We also thank the German Maritime and Hydrographic Agency for supporting us with AIS data taken in
Bremerhaven, Hamburg and Kiel.
**Author contribution**
VM developed the idea, designed and supervised the study, evaluated part of the model results, prepared the
manuscript and wrote most of the text. MQ co-designed the study, wrote most of the text about the meteorological
situation and provided interpretations of the meteorology-chemistry interactions. JAA helped in designing the
study, performed CMAQ model runs and provided code for the emission data preparation. RB developed the
Lockdown Adjustment Factors, extrapolated emission data, and wrote the section about the emission data. LF



performed CMAQ model runs, evaluated CMAQ model results and observation data and provided most of the
plots. RP performed COSMO model runs, provided information for the meteorological data interpretation, wrote
the text about the COSMO setup and part of the text about the meteorological situation, and analysed COSMO
model results. JF developed emission extrapolation factors, and provided interpretation of the observational data.
DS analysed AIS data and calculated ship emission LAFs. EML collected and analysed observational data and
provided data interpretation. MR helped in designing the study, analysed and interpreted observational data for
suburban stations. RW collected data on aviation emissions, provided LAFs for aviation and contributed to the
discussion of the results
**Appendix A**
**A1 Emission data**
**Table A1: Overview on available emission reduction information for countries in the investigated domain during the**
**lockdown applied in this study**

| Country or Ocean Area | A_PublicPower | B_Industry | F_RoadTransport | G_Shipping | G_Shipping_Inland | H_Aviation |
|---|---|---|---|---|---|---|
| Albania | | | | | x | x |
| Austria | x | x | x | | x | x |
| Baltic Sea | | | | x | | |
| Belarus | | | x | | x | x |
| Belgium | x | x | x | | x | x |
| Bosnia and Herzegowina | x | | x | | x | x |
| Bulgaria | x | x | x | | x | x |
| Croatia | x | x | x | | x | x |
| Cyprus | x | | | | x | x |
| Czech Republic | x | x | x | | x | x |
| Denmark | x | x | x | | x | x |
| Estonia | x | | x | | x | x |
| Finland | x | x | x | | x | x |
| France | x | x | x | | x | x |
| Germany | x | x | x | | x | x |
| Greece | x | | x | | x | x |
| Hungary | x | x | x | | x | x |
| Iceland | | | | | x | x |
| Ireland | x | | x | | x | x |
| Italy | x | x | x | | x | x |
| Latvia | x | | x | | x | x |





| | | | | | | |
|---|---|---|---|---|---|---|
| Liechtenstein | | | x | | x | |
| Lithuania | x | | x | | x | x |
| Luxembourg | x | x | x | | x | x |
| Malta | x | | x | | x | x |
| Moldova | | | x | | x | x |
| Montenegro | x | | | | x | x |
| Netherlands | x | x | x | | x | x |
| North Macedonia | x | | x | | x | x |
| North Sea | | | | x | | |
| Norway | x | | x | | x | x |
| Poland | x | x | x | | x | x |
| Portugal | x | x | x | | x | x |
| Romania | x | x | x | | x | x |
| Russia | | | x | | x | x |
| Serbia | x | | x | | x | x |
| Slovakia | x | x | x | | x | x |
| Slovenia | x | x | x | | x | x |
| Spain | x | x | x | | x | x |
| Sweden | x | | x | | x | x |
| Switzerland | x | | x | | x | x |
| Turkey | x | | x | | x | x |
| United Kingdom | x | x | x | | x | x |
| Ukraine | | | x | | x | x |

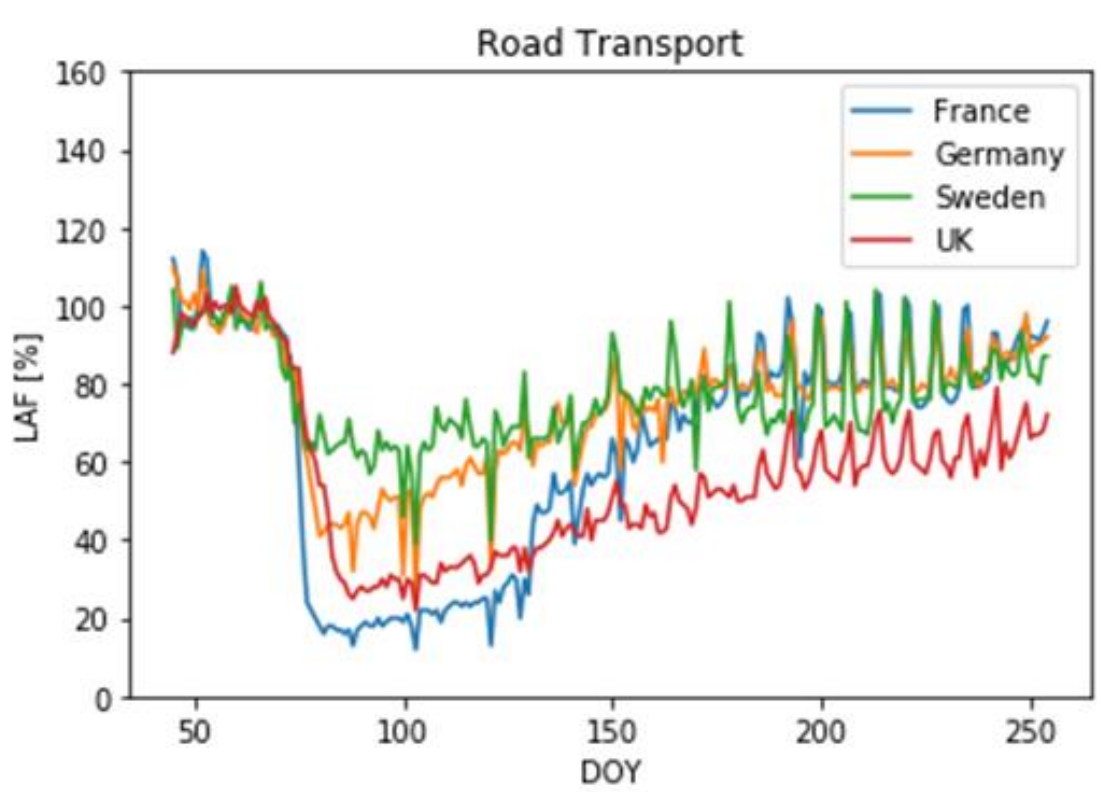

**Figure A1: Daily values for Lockdown Adjustment Factors (in %) for the sector F_RoadTransport based on transit data from the Google Mobility Reports.**

818

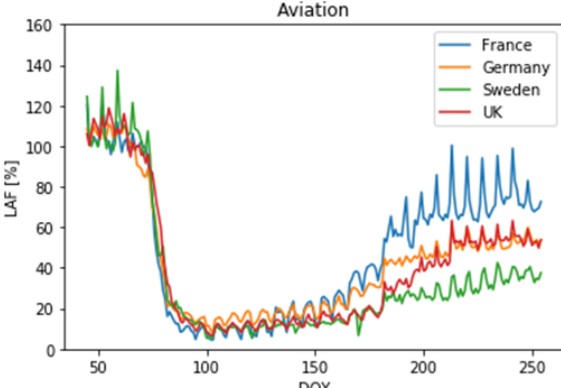

819

**Figure A2: Daily values for Lockdown Adjustment Factors (in %) for the sector H_Aviation based on Eurocontrol data.**

**A2 Meteorological situation**

**Table A2: GWL classification for the period 1 Februray 2020 – 31 May 2020**

| Date range | GWL |
|---|---|
| 01.02. - 02.02. | Cyclonic Westerly |
| 03.02. - 05.02. | Cyclonic North-Westerly |
| 06.02. - 08.02. | High over Central Europe |
| 09.02. - 12.02. | Cyclonic Westerly |
| 13.02. - 16.02. | Anticyclonic South-Westerly |
| 17.02. - 25.02. | Cyclonic Westerly |
| 26.02. - 28.02. | Cyclonic North-Westerly |
| 29.02. - 03.03. | Trough over Western Europe |
| 04.03. - 06.03. | South-Shifted Westerly |
| 07.03. - 09.03. | Maritime Westerly (Block E. Europe) |
| 10.03. - 12.03. | Cyclonic Westerly |
| 13.03. - 16.03. | Zonal Ridge across Central Europe |
| 17.03. - 20.03. | Anticyclonic Westerly |
| 21.03. - 26.03. | Scandinavian High Ridge C. Europe |
| 27.03. - 29.03. | Anticyclonic North-Easterly |
| 30.03. - 01.04. | Anticyclonic Northerly |
| 02.04. - 04.04. | Anticyclonic North-Westerly |
| 05.04. - 08.04. | Anticyclonic Southerly |
| 09.04. - 11.04. | High over Central Europe |
| 12.04. | undefined |
| 13.04. - 15.04. | High over the British Isles |
| 16.04. - 18.04. | Icelandic High Ridge C. Europe |
| 19.04. - 23.04. | High Scandinavia-Iceland Ridge C. Europe |
| 24.04. - 26.04. | Anticyclonic North-Westerly |





| 27.04. - 29.04. | South-Shifted Westerly |
|---|---|
| 30.04. - 02.05. | Cyclonic Westerly |
| 03.05. - 05.05. | Anticyclonic Northerly |
| 06.05. - 08.05. | High over Central Europe |
| 09.05. - 12.05. | Icelandic High Trough C. Europe |
| 13.05. - 15.05. | Anticyclonic North-Westerly |
| 16.05. - 18.05. | Zonal Ridge across Central Europe |
| 19.05. - 23.05. | High over Central Europe |
| 24.05. - 27.05. | Anticyclonic Northerly |
| 28.05. - 30.05. | Anticyclonic North-Easterly |
| 31.05. - 02.06. | High Scandinavia-Iceland Ridge C. Europe |


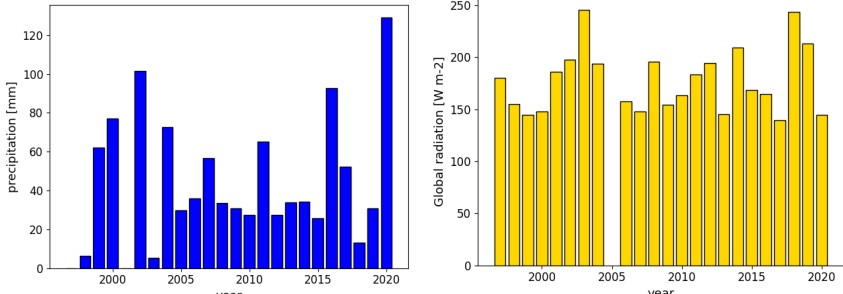

**Figure A3: Time series of the monthly accumulated precipitation and mean solar irradiance between 10 and 14 UTC**
**at the Wettermast Hamburg for February from 1997-2020.**

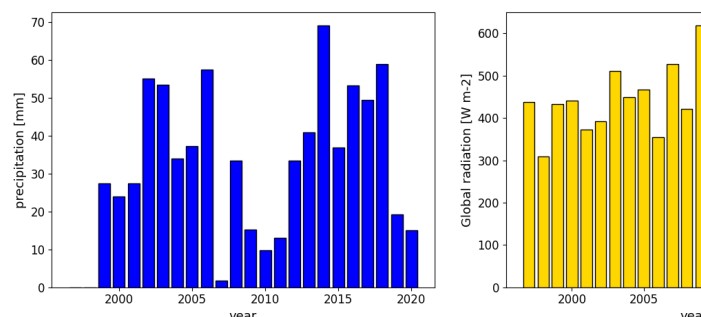

**Figure A4: Time series of the monthly accumulated precipitation and mean solar irradiance between 10 and 14 UTC**
**at the Wettermast Hamburg for April from 1997-2020.**





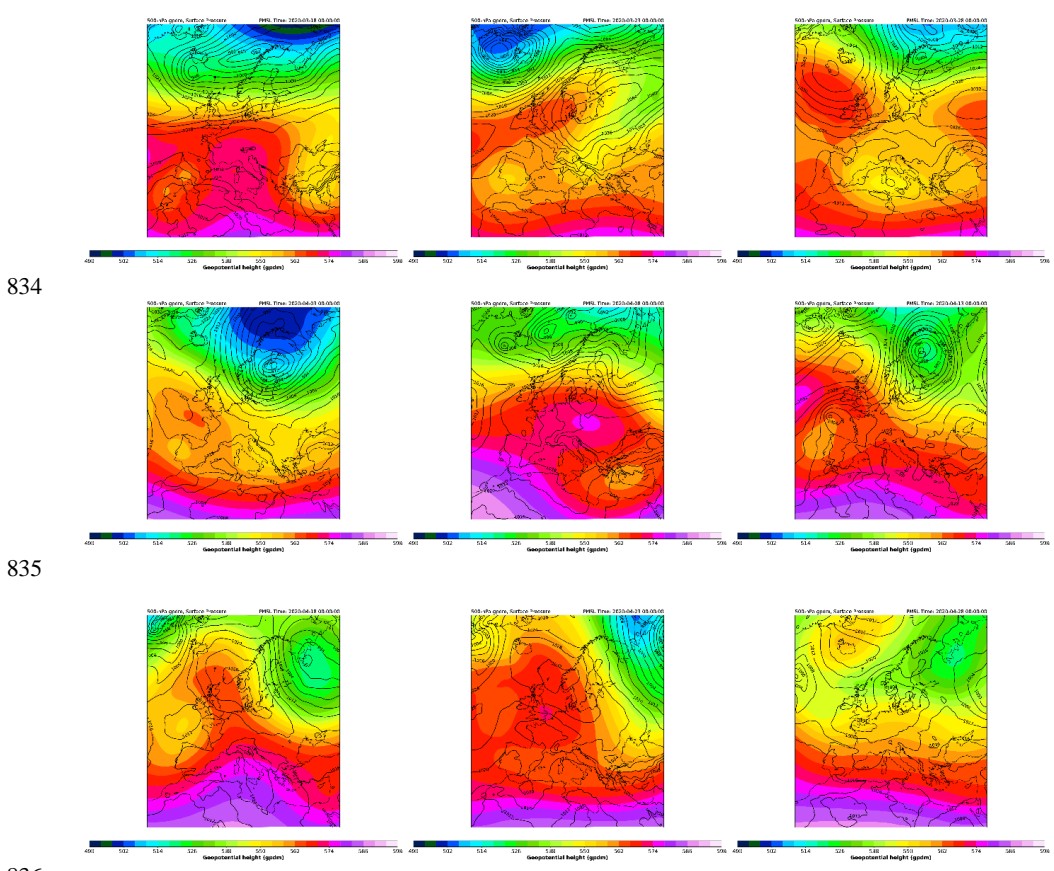




**Figure A5: 500 hPa geopotential heights (in gpdm) and surface pressure (in hPa) for 4-days time segments in March and April 2020 according to the COSMO simulations. The geopotential heights are averaged over 4 days, displayed surface pressure distributions are representative snap shots within those time segments.**


**A3 COVID-19 lockdown effects**


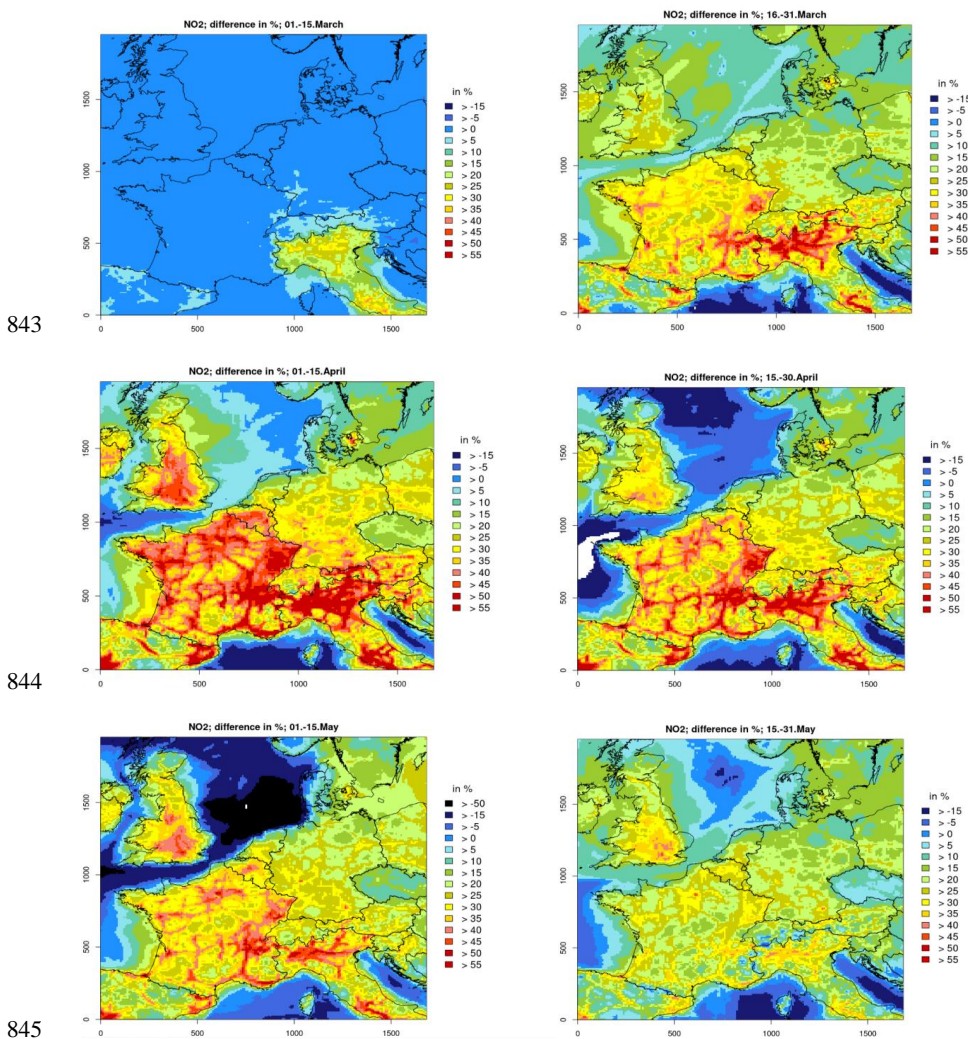




**Figure A6: CMAQ results for relative NO2 concentrations reductions due to lockdown measures (noCOV – COV run)**
**in Central Europe between 1 March and 31 May 2020 in half-monthly intervals; positive values denote reductions.**





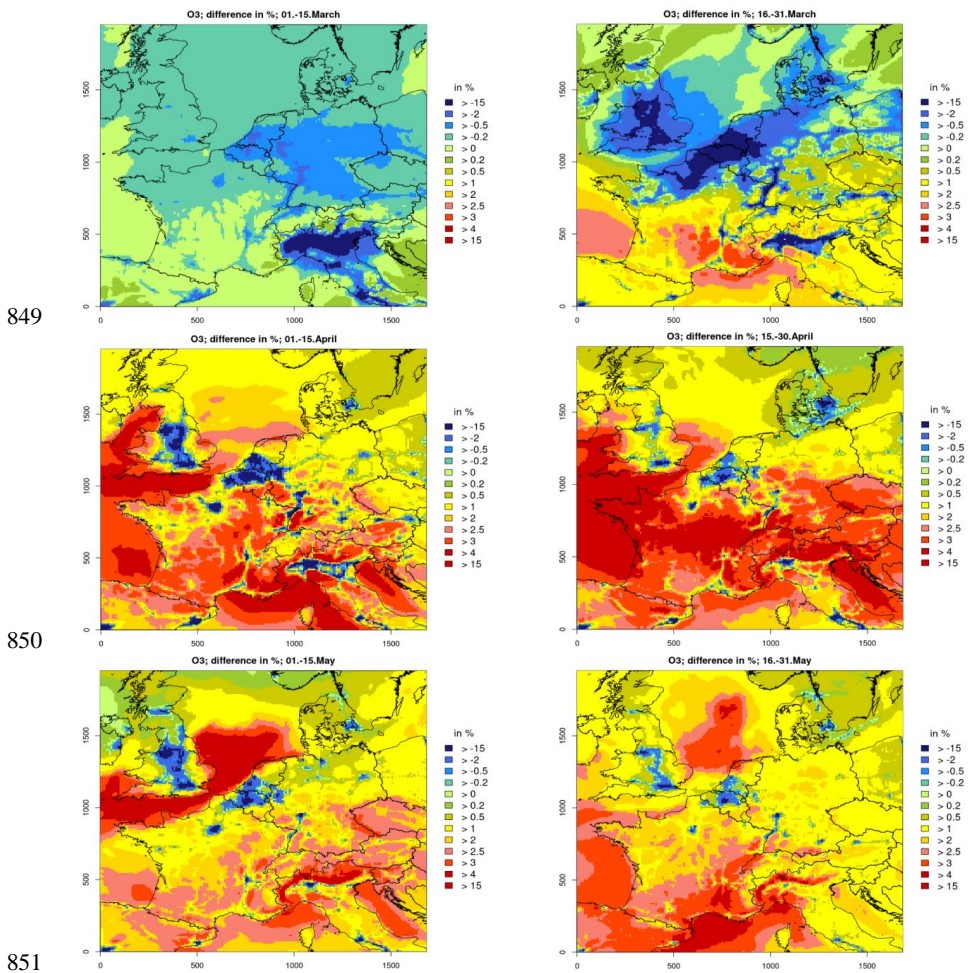




**Figure A7: CMAQ results for relative O$_3$ concentrations reductions due to lockdown measures (noCOV – COV run)**
**in Central Europe between 1 March and 31 May 2020 in half-monthly intervals; positive values denote reductions.**

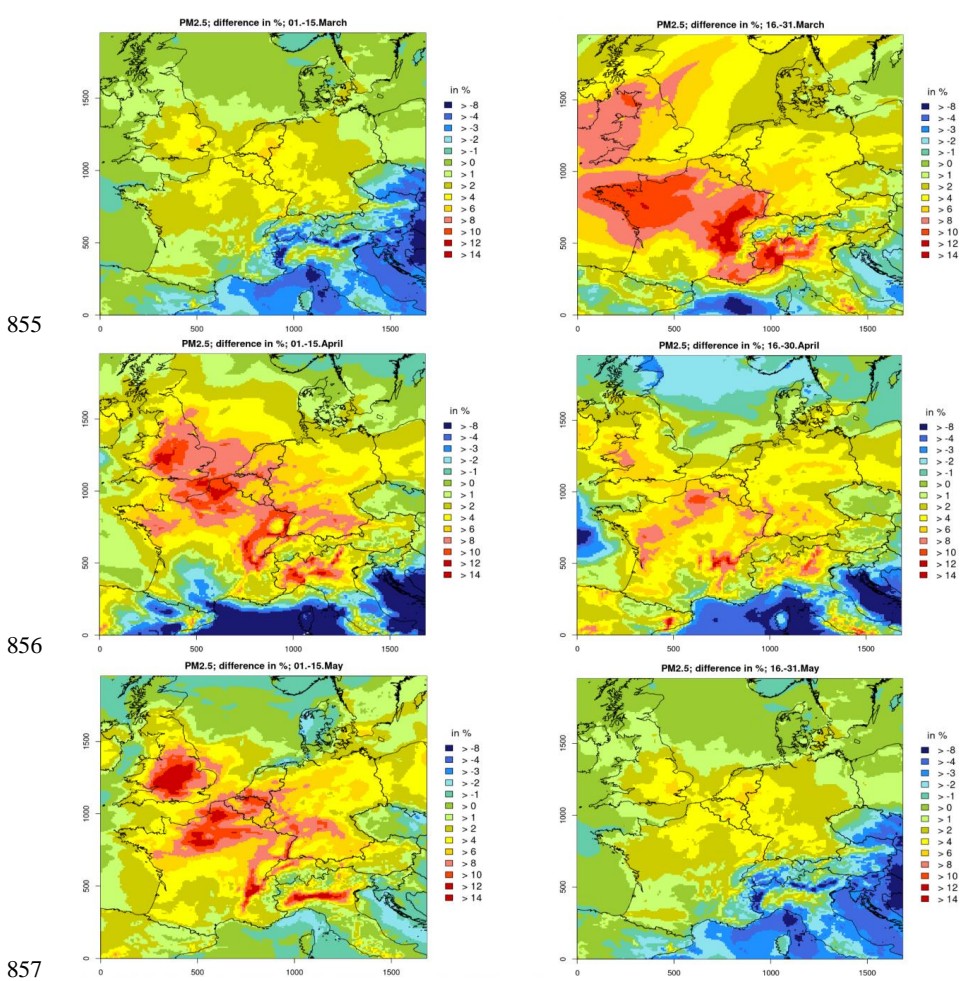




**Figure A8: CMAQ results for relative PM2.5 concentrations reductions due to lockdown measures (noCOV – COV run) in Central Europe between 1 March and 31 May 2020 in half-monthly intervals; positive values denote reductions.**



**A4 Discussion**




**Figure A9: Geopotential height at 500 hPa (in gpdm, isolines) and windspeed at 850 hPa (in m/s, color code): Differences between 2020 and 2018 (left column) and 2020 and 2016 (right column) for the half month-periods 16 macrh – 31 March (top), 1 April – 15 April (middle) and 16 April – 30 April (bottom).**






**Figure A10: Solar irradiance (in W/m², color code): Differences between 2020 and 2018 (left column) and 2020 and 2016 (right column) for the half month-periods 16 March – 31 March (top), 1 April – 15 April (middle) and 16 April – 30 April (bottom).**








**Figure A11: Accumulated precipitation (in mm, color code): Differences between 2020 and 2018 (left column) and 2020**
**and 2016 (right column) for the half month-periods 16 March – 31 March (top), 1 April – 15 April (middle) and 16**
**April – 30 April (bottom).**





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
