# Peer review of "The role of emission reductions and the meteorological"

_Atmospheric Chemistry and Physics, 2021_

## Author Comment (AC2)

Development of shipping emissions and transport work in the Baltic Sea according to HELCOM.
**https://portal.helcom.fi/meetings/MARITIME%2020-2020-787/Documents/Presentation%204_Ship%20emissions%20in%20the%20Baltic%20Sea%20area%20 2006%20-%202019.pdf**

---

## Author Response (AR1)

Point by point responses to the comments of Referee 1

We thank the referee for the valuable and very constructive comments that helped improving the manuscript considerably. All comments are repeated in this point-by-point answer of the authors. Reviewer comments are written in *Italics* while author's responses are given in **Bold**.

Comments Referee 1

*Review of "The role of emission reductions and the meteorological situation for air quality improvements during the COVID-19 lockdown period in Central Europe" by Matthias et al.*

*This manuscript provides a comprehensive, methodological analysis of the individual and combined effects of COVID-related emission changes and meteorological variability on air quality over Central Europe during the core period of the COVID-19 lockdown in 2020. The study design is thoughtful and sound. The results provide a valuable contribution to the rapidly-growing set of studies investigating this topic, especially by highlighting the complex interactions between meteorology and emissions for key pollutants and cautioning against attributing observed concentration changes directly to changes in emissions without performing an in-depth analysis of potential confounding factors.*

*The manuscript is generally well written and organized.*

*The introduction section could potentially be shortened by either eliminating or reducing the summary of results from previous studies.*
**It was shortened**

*My only major comment is to consider adding analysis for modelled PM2.5 species to provide additional context on how changes in total PM2.5 are driven by how different processes (emission changes vs. meteorology) affect individual PM2.5 components (e.g. primary vs. secondary, inorganic vs. organic).*
**We added results for ammonium, nitrate, and sulphate in the appendix and we briefly discuss this in the context of interactions between these inorganic aerosol components when precursor emissions (here NOx emissions) are strongly reduced. We refrained from discussing results for EC and OC in order to not further extend the paper that is already quite long.**

Specific comments:

*Page 1, line 22: remove comma after "both"*
**done**

*Page 2, line 46: suggest moving "also" after "weather conditions"*
**done**

*Page 2, line 48: To my knowledge, Goldberg et al. (2020) is a notable exception to this statement and might be cited here: Goldberg, D. L., Anenberg, S. C., Griffin, D., McLinden, C. A., Lu, Z., & Streets, D. G. (2020). Disentangling the impact of the COVID-19 lockdowns on urban NO2 from natural variability. Geophysical Research Letters, 47, e2020GL089269. https://doi.org/10.1029/2020GL089269*
**The statement does not imply that there are no studies about the interaction of the meteorological situation and the lockdown emission reduction, but that they care rare. We included the publication by Goldberg et al. later in the introduction when we describe previous studies.**

*Page 2, lines 49 – 54: This section seems to summarize results obtained later in the paper without explicitly saying so, but without providing any separate reference, either. I suggest either providing a reference or removing it from this portion of the manuscript.*
**We shortened this paragraph and included a statement about meteorological influences on photochemistry in the remaining text.**

*Page 4, line 126: were the COVID-19 lockdown effects considered in the IFS-CAMS fields used as boundary conditions? If not, does this introduce an additional level of uncertainty into the analysis, especially as it relates to the role of meteorology and longer-range air mass transport?*
**IFS-CAMS fields do not consider lockdown effects. We added a sentence about this on page 4.**
**Consequently, effects of lockdown measures outside Europe, e.g. in North America and Africa, on**

intercontinental transport are not considered in our simulations. However, the simulations consider emissions changes in entire Europe, while the evaluation is performed for Central Europe only. This setup already consider medium range transport inside Europe and reduces effects of intercontinental pollutant transport. In addition, intercontinental transport will not play a major role during the major lockdown period because the Großwetterlage with a blocking high pressure system in Central Europe did not favour this. In conclusion, we believe that the uncertainties caused by neglecting lockdown measures outside Europe are much lower compared to the inherent model uncertainties, which are now given in section 4.3 (former section 6.1).

*Page 5, lines 145-146: suggest moving "best" from the end of the sentence to before "reproduces"*
**done**

*Page 5, line 169: can you please provide a reference for the NMVOC split profiles used in this analysis?*
**The data was provided by Jeroen Kuenen from TNO in a personal communication. There is currently no reference for the data available, which is why we cited it as "personal communication" and added an acknowledgement to Jeroen Kuenen.**

*Page 6, line 187: add comma after "time series data"*
**done**

*Page 6, lines 195 – 196: What was the rationale for not assuming any changes in shipping emissions between 2016 and 2020?*
**This is based on data for the Baltic Sea published by HELCOM and the Finnish Transport and Communications Agency ( see https://portal.helcom.fi/meetings/MARITIME%2020-2020-787/Documents/Presentation%204_Ship%20emissions%20in%20the%20Baltic%20Sea%20area%20 2006%20-%202019.pdf) that shows a stable or even decreasing shipping emissions in the Baltic Sea when only IMO registered ships (i.e. bigger ships) are considered. We conclude from this that also in the North Sea shipping emissions will most likely not show significant changes between 2016 and 2020.**
**In order to keep the description of the basic emission construction for 2020 concise, we do not explain this further in subsection 3.1**

[Figure]

[Figure]

*Page 7, lines 224 – 228: You may want to state upfront that this approach cannot distinguish between passenger cars and trucks which likely had very different activity changes resulting from the lockdown. This limitation is discussed in Section 6.2 but in my opinion should be mentioned here.*
**We added that vehicle types cannot be distinguished.**

*Page 7, line 237: most readers likely aren't familiar with the term RoRo for certain types of ferries, please define or spell out.*
**We now explain this in the text, Roll-on/Roll-off**

*Page 11, lines 286: suggest changing "... exceptional weather, what is assumed" to "exceptional weather that is assumed"*
**done**

*Page 11, line 301: change "supplemented" to "supplemental"*
**done**

*Page 12, line 327: remove comma after "meteorological fields"*
**done**

*Page 13, line 372: suggest moving "also" from before "advected pollutants" to after "meteorological conditions"*
**done**

*Page 13, line 373: add comma before "time series"*
**done**

*Page 14, line 386: add comma before "time series"*
**done**

*Page 20, lines 487 – 497 and Figure 12: recommend adding analysis and discussion for key PM2.5 species (sulfate, nitrate, ammonium, EC, OC) – see major comment above.*
**We analysed the main PM components sulphate, nitrate, and ammonium which contribute more than 2/3 of the total modelled PM2.5. We added a paragraph about the results in section 5.1 below the paragraph about PM2.5 concentrations. Figures that show the temporal development of the changes in sulphate, nitrate, and ammonium are given in the appendix. We refrained from extending the discussion for more details and other PM2.5 components, because the manuscript is already very long. In addition, model results about BC and OC are less reliable than those about secondary inorganics, as previous model intercomparison studies have shown. This is because BC, and also NMVOC emissions are still quite uncertain. In addition, SOA formation is usually underestimated in CMAQ model results.**

*Page 22, line 511: suggest replacing "observed" with "simulated" to avoid confusion*
**done**

*Page 23, line 531: remove comma after "both"*
**done**

*Page 23, line 539: remove comma before "only"*
**done**

*Page 26, lines 604 – 605: Differences between observations and model simulations likely also are caused by other errors in the modeling system (uncertainties in simulated meteorological fields, chemistry, deposition, base emission inventory, etc.), not only uncertainties in representing the lockdown effects. Suggest reconsidering this statement.*
**We added two sentences about typical model uncertainties.**

*Page 30, line 652: change "(Bauwens et al., 2020)" to "Bauwens et al., (2020)"*
**done**

*Page 30, line 661: remove comma after "selected"*
**done**

*Page 30, line 670: remove comma after "constellation"*
**done**

*Page 31, line 692: remove comma after "conditions"*
**done**

*Page 31, line 718: PM2.5 is both primary and secondary. My suggestion of adding analysis for PM2.5 components would potentially shed light on which portions of the PM2.5 changes are more sensitive to emissions changes vs. meteorology.*
**This is a very interesting investigation, but we think that it is not possible to do this in detail in this paper. Inorganic PM is now briefly discussed in section 5.1 and a number of figures was added in the appendix. A further discussion as the reviewer suggests would extend the entire paper which is already quite long. We consider to discuss changes in PM components in separate study based on the same model runs.**

*Page 33, line 769: remove comma before "only"*
**done**

Point by point responses to the comments of Referee 3

We thank the referee for the valuable and very constructive comments that helped improving the manuscript considerably. All comments are repeated in this point-by-point answer of the authors. Reviewer comments are written in *Italics* while author's responses are given in **Bold**.

Comment Referee 3

*This manuscript aims to assess the roles of emission reduction due to Covid-19 lockdown and meteorology in air quality during January-June 2020 in Central Europe. It first developed detailed emission inventories for this period based on the previous year's emission data and various activity data during the lockdown, and then examined meteorological conditions during the first half of 2020 in Central Europe and compared air quality data collected from 6 sites in the above period with those from previous years. Then a chemical transport model was used to simulate the air quality in 2020 and sensitivity runs were conducted to assess the impact of emission reductions during the lockdown and the effect of meteorology change to year 2016 and 2018. The key findings include a large reduction in NO2 concentration due to city lockdown, increase or decrease in O3, and smaller reduction in PM2.5 concentrations. It also demonstrated the importance meteorology in modulating the air quality.*

*The manuscript adds useful information to the large body of literatures on air quality during Covid-19 lockdown and the complex interplay of emission, meteorology and atmospheric chemistry. The methods are reasonable, though not particularly new. My main comment is that there are few discussions to convey new insights/findings of this study in comparison with numerous studies by other investigators on the issue.*
**See the answers to the follow-up comments below.**

*As shown in the introduction, there have been quite a lot of studies on the response of air quality in Europe (also see two more recent papers), what is new in the methodology adopted in the present work?*
**Compared to earlier more limited studies on the same topic, this investigation aims at eliminating some of the restrictions seen there. In the following aspects, this study differs from previous approaches:**
- **A very detailed emission construction for bas case (noCOV) emissions in 2020 considering changes between 2016 and 2020 on sectoral national level.**
- **A very detailed construction of lockdown emissions, also compared to Dubois (see our response to another comment later in this document) and Guevara et al. (2020), who considered a shorter period.**
- **A comprehensive view on the effects of emission reductions and the meteorological situation in entire Central Europe. Similar studies are limited to one country or smaller regions (e.g. Velders et al. 2021) or shorter time periods (Menut et al., 2020).**
- **Simulations with other meteorological data from previous years but the same emission data revealing the impact that meteorological conditions can have.**

*And what are new findings of the analysis?*
**Our main new findings are the following:**
- **Lockdown emission reduction effects on ozone vary in time and place from concentration increases to decreases. In May and June, ozone increases can only be seen in cities and very limited areas with high NOx emissions**
- **NOx emission reductions during the lockdown resulted in decreased nitrate concentrations, but slightly increased sulphate concentrations because of more available ammonia.**
- **The meteorological situation in April 2020 was far more important for low PM concentrations in Northern Europe than lockdown emission reductions.**
- **Secondary pollutants like ozone and PM show larger variations caused by meteorological influences than by lockdown emission reductions in Europe.**
- **Observations only are not sufficient to say something about the lockdown effects on concentrations.**

*I am impressed by the careful development of the emission inventories for Central Europe for January-June 2020, how does it compare to another recent inventory (Doumbia et al. 2021)?*
**There are many similarities between the inventories and the methods that were used to derive Lockdown Adjustment Factors, e.g. the use of google mobility data for changes in road traffic. The data set presented by Doumbia et al. covers all continents which is why sometimes different data sets are combined or data sets are used that cover more regions than Europe. This study only uses data sets to derive LAFs that are available for almost all European countries. Some of them have higher temporal resolution than the data**

used by Doumbia (e.g. data for aviation and shipping). Doumbia et al. estimate higher emissions from the residential sector based on google residential data while we left this sector unchanged. The reasoning behind this that the heating demand is most likely not significantly modified when more people stay at home compared to the case when they go to work.

One main difference in the 2020 emission dada sets is that we extrapolated 2016 CAMS emissions for Europe to the year 2020 to derive the emissions for the noCOV case. Estimated total emission reductions for Europe from both models are quite close (25 -30% decrease in NOx in April.

We added a few sentences about the relation between the inventories in section 6.2.

*Another comment is about the organization of this paper. It is currently very lengthy, and I think it can be shortened to highlight the novel parts of this study.*

We reorganized the paper by moving subsection 6.1 upwards at the end of section 4 (see our answer to the next comment) and shortened it. We removed the data about the average concentration changes at the measurement stations in the first 6 months of the year 2020 because it does not add much information to what is described at the end of section 5.1. We moved the text about the differences between the meteorological situation in 2016 and 2018 to the appendix, because this is sufficiently described in subsection 5.2.

Although information about the modelled PM species was added, the main text of the paper (excluding appendix and references) is now shorter by 500 words.

*I also suggest moving the model validation part presented in a later part to the earlier part before showing the model results.*

We moved subsection 6.1 upwards at the end of section 4. It is now the new subsection 4.3 and the title was changed into "Comparison between model results and observations". Subsections in section 6 were modified accordingly and the numbering of the figures was adapted.

*The discussion section (Section 6) is a really part of the general results, with no in-depth discussion such as a comparison with other researchers' work and significance/implications of the results, which should be included.*

We moved subsection 6.1 to section 4 and present the comparison between model and observations there. We now put our results in perspective to previous studies, in particular for NO2 and O3. We also discuss interactions between inorganic PM components and uncertainties related to SOA modelling. We moved the part about differences between the meteorological situation in 2016 and 2018 to the appendix and thereby focused the discussion on the emission estimates and the modelled concentrations.

*The conclusion section is rather a summary of the results and is also unnecessarily long. It should be condensed and highlight the key findings.*

It was shortened at several places and now the key findings are more prominently presented. One sentence about lockdown effects on PM components was added.